# InPACT: a computational method for accurate characterization of intronic polyadenylation from RNA sequencing data

Xiaochuan Liu [1,7], Hao Chen[2,7], Zekun Li [1], Xiaoxiao Yang [1,3], Wen Jin [1,3], Yuting Wang[1,3], Jian Zheng [4], Long Li [4], Chenghao Xuan [2] ✉, Jiapei Yuan [5,6] ✉ & Yang Yang [1,3] ✉

Alternative polyadenylation can occur in introns, termed intronic polyadenylation (IPA), has been implicated in diverse biological processes and diseases, as it can produce noncoding transcripts or transcripts with truncated coding regions. However, a reliable method is required to accurately characterize IPA. Here, we propose a computational method called InPACT, which allows for the precise characterization of IPA from conventional RNA-seq data. InPACT successfully identifies numerous previously unannotated IPA transcripts in human cells, many of which are translated, as evidenced by ribosome profiling data. We have demonstrated that InPACT outperforms other methods in terms of IPA identification and quantification. Moreover, InPACT applied to monocyte activation reveals temporally coordinated IPA events. Further application on single-cell RNA-seq data of human fetal bone marrow reveals the expression of several IPA isoforms in a context-specific manner. Therefore, InPACT represents a powerful tool for the accurate characterization of IPA from RNA-seq data.

The process of mRNA precursor maturation through cleavage and polyadenylation at polyadenylation (polyA) sites is a critical step of post-transcriptional regulation[1]. Alternative cleavage and polyadenylation (APA) is recognized as an important post-transcriptional regulatory mechanism that generates multiple RNA transcripts from a single gene through the selection of various polyA sites. APA has been implicated in diverse biological processes, including immune response, stem cell differentiation, and cancer progression[2–7]. APA can be categorized as different types based on the genomic location of polyA sites, such as 3' untranslated region (UTR) APA and intronic APA

(IPA)[8,9]. The 3'UTR APA events arise in the 3'-most exons, leading to the generation of tandem 3'UTR isoforms without altering the protein-coding sequence, while IPA events take place within introns, giving rise to the generation of alternative last exon isoforms[8,9]. The regulatory role of 3'UTR APA in modulating gene expression has been demonstrated through its impact on mRNA stability, localization, and translation efficiency[10–12]. In contrast, IPA can not only give rise to isoforms with distinct 3'UTRs, but also result in the production of either noncoding transcripts or truncated protein-coding transcripts with the loss of C-terminal domains in the protein product[2,3,5,13–15].

[1]The Province and Ministry Co-sponsored Collaborative Innovation Center for Medical Epigenetics, Tianjin Key Laboratory of Inflammatory Biology, The Second Hospital of Tianjin Medical University, Department of Bioinformatics, School of Basic Medical Sciences, Tianjin Medical University, Tianjin 300070, China. [2]Department of Biochemistry and Molecular Biology, School of Basic Medical Sciences, Tianjin Medical University, Tianjin 300070, China. [3]Department of Pharmacology, School of Basic Medical Sciences, Tianjin Medical University, Tianjin 300070, China. [4]Department of Immunology, School of Basic Medical Sciences, Tianjin Medical University, Tianjin 300070, China. [5]State Key Laboratory of Experimental Hematology, National Clinical Research Center for Blood Diseases, Haihe Laboratory of Cell Ecosystem, Institute of Hematology and Blood Diseases Hospital, Chinese Academy of Medical Sciences and Peking Union Medical College, Tianjin 300020, China. [6]Tianjin Institutes of Health Science, Tianjin 301600, China. [7]These authors contributed equally: Xiaochuan Liu, Hao Chen. ✉e-mail: chenghaoxuan@tmu.edu.cn; yuanjiapei@ihcams.ac.cn; yy@tmu.edu.cn

The IPA events can be categorized into two distinct groups: composite IPA events and skipped IPA events. Composite events involve the conversion of an internal exon into a 3′ terminal exon, whereas skipped events involve the utilization of a 3′ terminal exon that would be otherwise skipped[9]. The expression of IPA isoforms is demonstrated to exhibit a cell-type-specific manner, as evidenced by the immunoglobulin M heavy chain (IGHM) locus. Specifically, mature B cells produce full-length IGHM isoforms, whereas plasma cells generate IPA isoforms that result in the loss of the transmembrane domain and the secretion of IgM antibodies[3]. Recent investigations have illuminated the biological significance of IPA. In the context of B cell leukemia, aberrant IPA events have been demonstrated to generate truncated proteins that deactivate tumor suppressor genes, such as *DICER*, *FOXN3*, and *MGA*[2]. Furthermore, atypical IPA events have been observed to be augmented in solid tumors, including *TSC1* in lung cancer and *MAGI3* in breast cancer[13,14]. In addition to generating truncated proteins, IPA can also alter the 3′UTR content, thereby influencing the subcellular localization of RNA molecules. Taliaferro et al. have examined RNA subcellular localization at the isoform level and found that gene-distal alternative last exon isoforms preferentially localize to neurites[15]. These findings collectively suggest that many IPA isoforms may play an unanticipated role in various biological processes and pathological conditions.

Despite the crucial role of IPA in various biological and pathological processes, its precise genome annotations and biological significance remain incompletely understood. While a number of high-throughput sequencing techniques, such as A-seq, 3P-seq, 3′READS, PAS-seq, and PolyA-seq, have been developed to directly sequence 3′ ends of RNAs and detect polyA sites, they have not been widely utilized and the available data remain relatively scarce, impeding further investigation[16–20]. In contrast, RNA-seq has been extensively utilized in various samples and conditions, providing an opportunity to characterize APA. Although several computational tools, including MISO, QAPA, and LABRAT, enable APA quantification, they do not possess the capability to identify novel IPA isoforms[21–23]. Furthermore, several methods such as DaPars, Aptardi, APAtrap and TAPAS have been specifically designed for the identification of and quantification of APA based on RNA-seq data, but are limited to the analysis of 3′UTR APA[24–27]. Recently, a specialized tool called IPAFinder has been proposed to identify IPA sites and analyze the dynamic regulation of IPA using RNA-seq data[14]. However, IPAFinder relies on read coverage fluctuations, which necessitates adequate sequencing depth, and the precision of identified IPA sites raises concerns regarding subsequent analysis[14]. Additionally, a recently published method called APAIQ, which leverages the combined effect of DNA sequence and RNA-seq read coverage, can also be employed to identify IPA sites[28]. Nevertheless, APAIQ is not specifically designed for IPA analysis and therefore cannot assemble IPA isoforms and cannot differentiate between skipped and composite IPA events. Consequently, the precise characterization of IPA from conventional RNA-seq remains a challenge, impeding the progress towards a comprehensive understanding of IPA.

Here, we present a new computational method, InPACT (Intronic PolyAdenylation Characterization Tool), which incorporates a sequence module and a read module to enable precise sample-wise characterization of IPA using conventional RNA-seq data (Fig. 1). InPACT can reproducibly and accurately identify IPA sites and reconstruct IPA isoforms. 3′-Rapid Amplification of cDNA Ends (3′-RACE) experiments have been conducted to validate the presence of several predicted IPA sites. We show that InPACT-predicted IPA isoforms are sufficiently stable to undergo translation, as evidenced by ribosome profiling data. Demonstrating the effectiveness of our method, we show that InPACT outperforms IPAFinder in identifying and quantifying IPA with 3′-end sequencing data, long-read sequencing data and simulated RNA-seq data as benchmarks. By leveraging InPACT, we have profiled and determined the dynamics of novel IPA events in monocyte activation. Furthermore, we illustrate the potential of InPACT in the investigation of IPA using human fetal bone marrow single-cell RNA-seq data, thereby enabling the characterization of cell-type-specific IPA events. Collectively, our results underscore the potential of InPACT to facilitate the detection and characterization of IPA from conventional RNA-seq data, thereby enabling a more comprehensive understanding of IPA in diverse biological processes and pathological conditions.

## Results

### InPACT design

The InPACT was designed to effectively identify and quantify IPA events via the examination of contextual sequence patterns and RNA-seq reads alignment. Notably, the modular structure of InPACT includes a sequence module that utilizes a convolutional neural network (CNN) to scan for all potential polyA sites within genomic regions annotated as introns, as well as a read module that employs a sample-specific classifier trained on features that characterize the alignment of sequencing reads generated from single- or paired-end RNA-seq. The methodology and architecture of InPACT are depicted in Fig. 1.

Briefly, the sequence module was designed to utilize a convolutional neural network architecture to learn from the genomic sequence surrounding annotated polyA sites (Fig. 1, see Methods). As the cis-regulatory elements are typically within 100 nt upstream and downstream of a polyA site[29], this module takes genomic sequence in 201 nt windows centered on target sites as input. These sequences were encoded using one-hot representation, resulting in a dimensionality of 4 × 201 (Fig. 1). With the human reference annotation of Refseq as an example, a CNN was trained to predict polyA sites, resulting in accurate predictions (Supplementary Fig. 1a). Then, we evaluated the performance of our trained CNN on three commonly used polyA databases: GENCODE, PolyA_DB 3, and PolyASite 2.0[30–32]. To ensure unbiased testing, we excluded any overlapping polyA sites used for model training and constructed separate sets of testing polyA sites. The results demonstrated that our CNN model effectively predicted the polyA sites in the aforementioned databases, achieving AUROC values of 0.954 for GENCODE, 0.920 for PolyA_DB 3, and 0.794 for PolyASite 2.0 (Supplementary Fig. 1b–d). When comparing our model's performance with three other deep-learning models (DeepPASS, APARENT, and DeepPASTA)[33–35], we found that our model's overall performance is either better or comparable (Supplementary Fig. 1b–d). Notably, all models exhibited lower performance on PolyASite 2.0 compared to other testing sets of polyA sites (Supplementary Fig. 1d). However, DeepPASS showed slightly better performance, which can be attributed to its training on a comprehensive set of polyA sites from multiple databases, including PolyASite 2.0[33]. Furthermore, we applied this model to scan for IPA sites across non-overlapped intronic regions annotated in the human genome, yielding candidate IPA sites. These findings demonstrated the potential utility of the sequence module in predicting polyA sites and identifying novel IPA sites.

Next, we designed a read module to accurately identify previously unannotated terminal exons located in non-overlapped intronic regions, which allows for the identification of IPA sites existing in the RNA-seq data from candidate IPA sites predicted from the sequence module (Fig. 1). This was accomplished by training a classifier specific to the RNA-seq data under analysis to distinguish terminal exons from background regions and internal exons, which was inspired by the previous study[36] (Supplementary Fig. 2, see Methods). Various features were constructed to characterize these regions from the alignment of RNA-seq data, including both spliced and unspliced reads surrounding the 5′ and 3′ boundaries, which was inspired by previous studies[9,24,36] (Fig. 1, Supplementary Fig. 3). The read module differs from previous methods by considering coverage patterns and alignment information, rather than merely read coverage. Two types of new terminal exons classified based on their structural composition were considered. The first type, referred to as composite terminal exon, spans

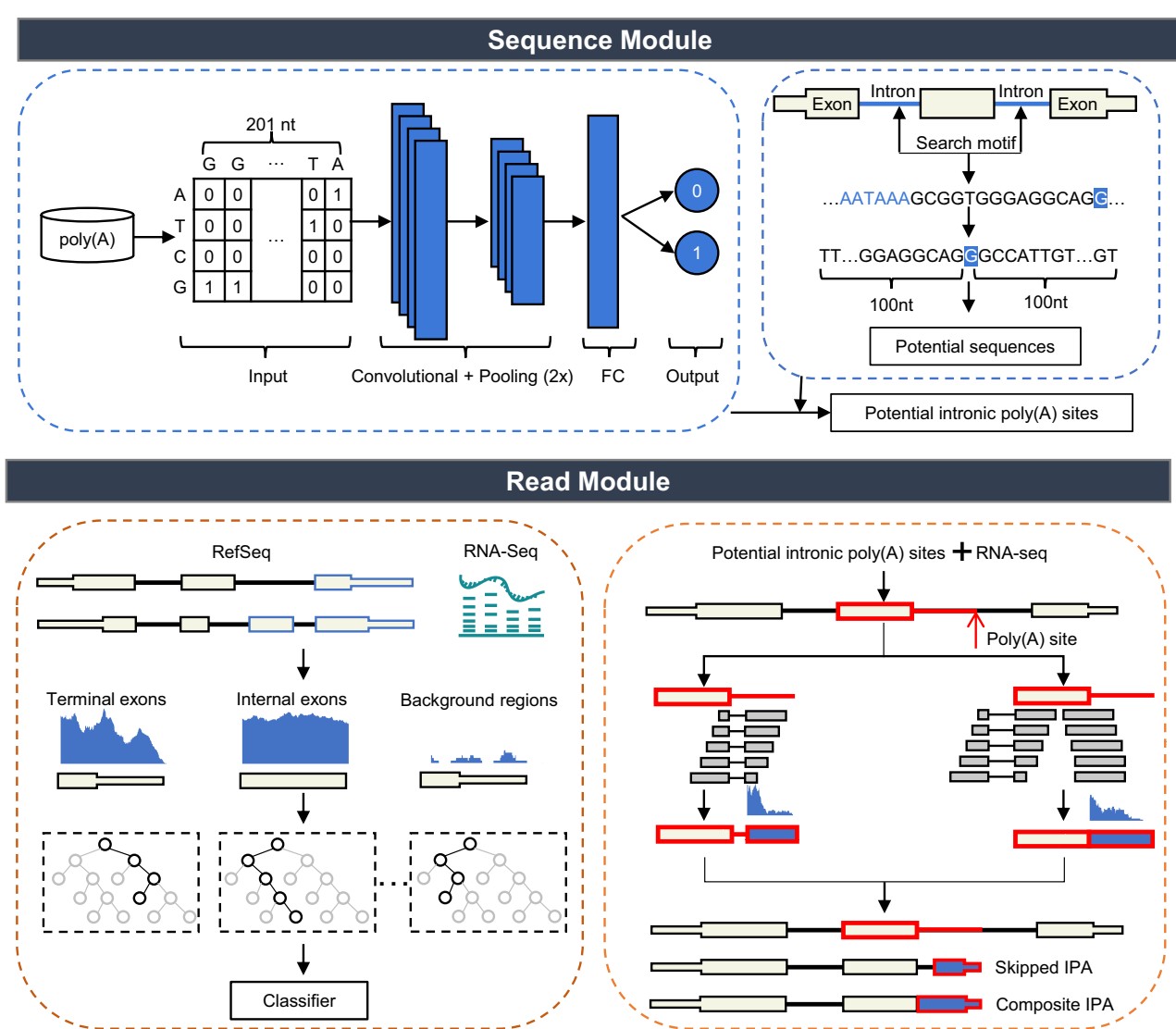

**Fig. 1 | Overview of the InPACT method.** Flow chart of the framework of InPACT incorporating two modules, namely the sequence module and the read module. The framework of the sequence module is illustrated in the upper panel, while the lower panel depicts the framework of the read module.

the entire sequence from the upstream donor splice site to the intronic polyA site (Supplementary Fig. 2). The second type, known as skipped terminal exon, results from the IPA event that introduces a novel exon ending at the intronic polyA site, necessitating the recognition of both an upstream donor splice site and a new acceptor splice site (Supplementary Fig. 2). To that end, InPACT can train separate models for skipped and composite candidates using corresponding feature combinations (Fig. 1, Supplementary Fig. 3). The accuracy of the models was tested with two published RNA-seq replicates of human embryonic kidney 293 (HEK293) cells[37] (Supplementary Data 1). The results showcased that both models effectively identified terminal exons in the testing set (Replicate 1: $AUROC_{skipped} = 0.999$, $AUROC_{composite} = 0.997$; Replicate 2: $AUROC_{skipped} = 0.998$, $AUROC_{composite} = 0.998$) (Supplementary Fig. 4). The performance of the models was evaluated using various metrics, including true negative rate, precision, F1-score, accuracy, true positive rate, and Matthew's correlation coefficient (Supplementary Fig. 4).

**InPACT reproducibly and accurately identifies intronic polyA sites**

Leveraging the two RNA-seq replicates of HEK293 cells, we conducted a comprehensive evaluation of the performance of InPACT

(Supplementary Data 1). 471 (319 skipped and 152 composite) and 393 (287 skipped and 106 composite) polyA sites that are novel with regard to the Refseq annotation were identified for those two replicates, respectively (Supplementary Fig. 5). As an example, a skipped IPA site of gene *ZNF771* was identified in HEK293 cells with a novel splice junction supported by many spliced reads (Fig. 2a). In addition, a composite IPA site of gene *TERF2* was identified in HEK293 cells, which was novel with respect to GENCODE annotation (Fig. 2b). In comparison, an overlap of approximately 50% (218 sites in total, with 182 skipped and 36 composite) was observed between two replicates (Supplementary Fig. 5a–c). The moderate consistency observed between the replicates may be ascribed to the relatively low coverage of those non-overlapped IPA isoforms (Supplementary Fig. 5d). To further validate the reliability of the IPA sites identified by InPACT, we conducted 3'-RACE experiments on 15 selected candidate IPA sites in HEK293 cells, including the IPA sites within gene *ZNF771* and *TERF2* (Fig. 2c, d, Supplementary Fig. 6, Supplementary Data 2). Remarkably, all candidate IPA sites were successfully confirmed using 3'-RACE in HEK293 cells. Specifically, the results showed that 10 out of the 15 candidate IPA sites were confirmed within 10 nt of their predicted positions, while the remaining 5 sites were confirmed within approximately 40 nt (Supplementary Data 2).

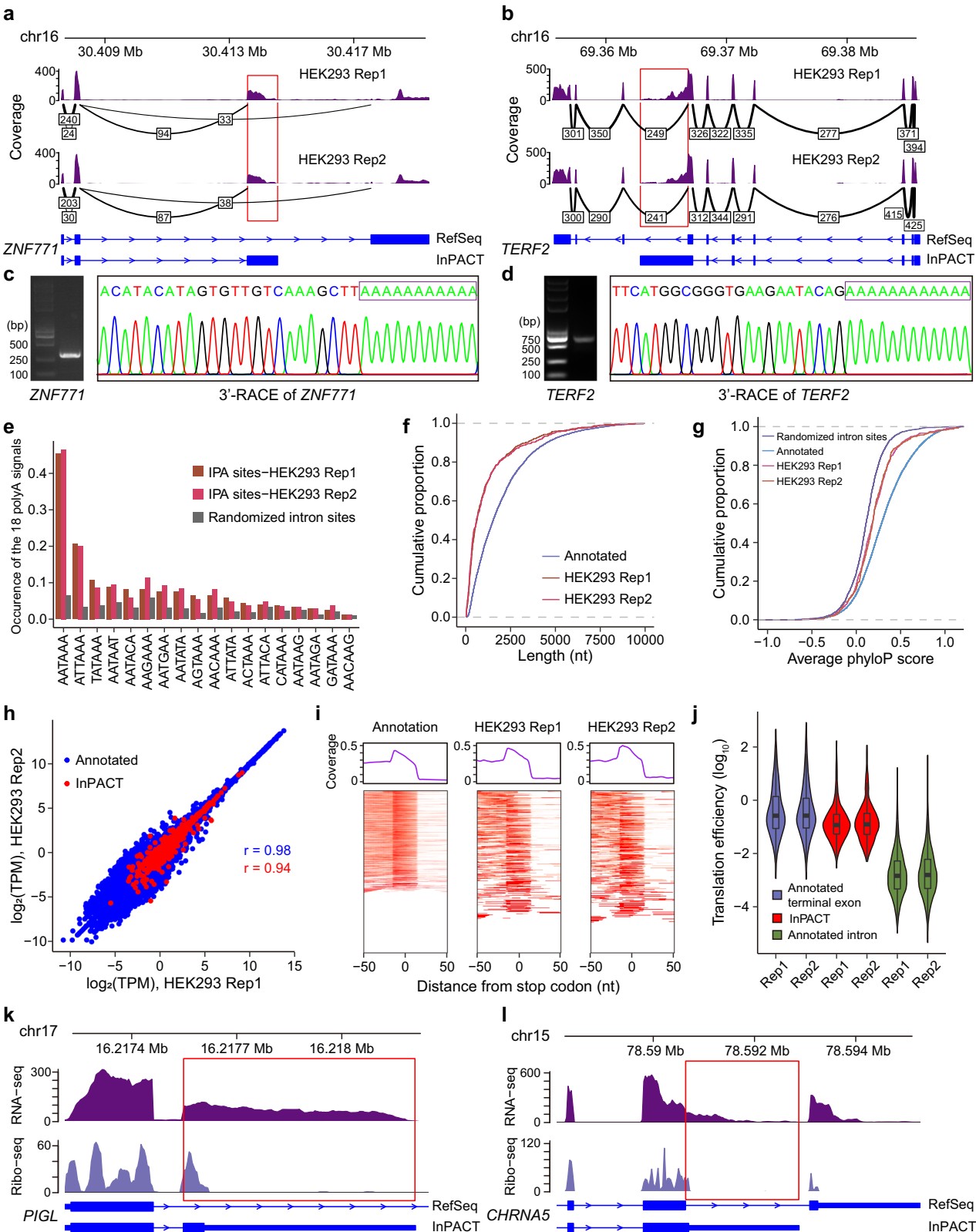

The polyA signals, which are typically located upstream of the polyA sites, are required for pre-mRNA cleavage and polyadenylation. The AAUAAA, AUUAAA, and related variants have been identified as canonical signals. To assess the reliability of InPACT, we examined the polyA signals and base compositions in the vicinity of IPA sites identified by InPACT. The results indicated that the canonical polyA signal (AAUAAA) was significantly enriched in each RNA-seq replicate, followed by AUUAAA (Fig. 2e). Additionally, the nucleotide profile surrounding the identified IPA sites was found to be similar to that obtained from Refseq annotated polyA sites. InPACT also has the capability to reconstruct intronic terminal exons corresponding to the IPA sites. The median lengths of intronic terminal exons identified by InPACT in two HEK293 RNA-seq replicates were found to be shorter compared to annotated ones (Fig. 2f). In addition, to assess their

**Fig. 2 | Identification of novel IPA sites in HEK293 cells using InPACT. a, b** Two examples of InPACT-identified IPA sites in HEK293 cells. The Sashimi plots depict the RNA-seq reads aligned to the *ZNF771* (**a**) and *TERF2* loci (**b**), with the annotation of Refseq isoforms and InPACT-identified novel IPA isoforms. The densities of RNA-seq reads from two RNA-seq replicates of HEK293 cells are shown in purple. Splice junctions are displayed as arcs connecting exons. The number of reads observed for each junction is indicated within the arc. **c, d** Experimental validation of InPACT-identified IPA sites in *ZNF771* (**c**) and *TERF2* (**d**) genes from HEK293 cells. The gel of 3'-Rapid Amplification of cDNA Ends (3'-RACE) experiments and Sanger sequencing results of the amplified transcripts by 3'-RACE experiments were depicted. Each experiment was repeated *n* = 3 times. **e** The bar plot depicts the frequencies of 18 known polyA signals detected in the regions of 60 nt upstream of the respective sites. The IPA sites identified from two RNA-seq replicates of HEK293 are compared with randomly selected genomic sites in introns. **f, g** The cumulative distribution curves of terminal exon length (**f**) and conservation scores (PhyloP) (**g**). The IPA terminal exons identified from two RNA-seq replicates of HEK293 are compared with annotated terminal exons or randomly selected genomic sites in introns. **h** The

scatter plot depicts the estimated expression levels of annotated isoforms (blue) and InPACT-identified novel IPA isoforms (red) from two RNA-seq replicates of HEK293. The Pearson correlation coefficients are indicated in the respective colors. **i** The plots depict the read coverage of ribo-seq reads around stop codons of annotated isoforms and IPA isoforms. The upper panels depict the average read coverage. The lower panels show the read coverage for each isoform using heat-maps. **j** The violin plot depicts translational efficiencies of annotated terminal exons (*n* = 1048 for rep1 and *n* = 925 for rep2), IPA terminal exons (*n* = 382 for rep1 and *n* = 312 for rep2) and introns (*n* = 9,268 for rep1 and *n* = 8284 for rep2). The center lines denote the median values with the boxes are bounded by the 25th and 75th percentiles. The whiskers extend to the maximum and minimum values within 1.5 times the interquartile range (IQR) from each end of the box. **k, l** Two examples of InPACT-identified IPA isoforms translated in HEK293 cells. The plots show the reads coverage of RNA-seq and ribo-seq in the locus for *PIGL* (**k**) and *CHRNA5* (**l**), with the annotation of Refseq isoforms and InPACT-identified IPA isoforms. The coding sequences and stop codons are illustrated in the annotation tracks.

evolutionary conservation, phyloP scores in regions ranging from 20 nt upstream and 20 nt downstream of polyA sites were computed. The identified IPA sites were observed to be moderately evolutionarily conserved, albeit a lesser extent than annotated polyA sites (Fig. 2g). These results collectively supported the authenticity of IPA sites identified by InPACT.

Moreover, InPACT possesses the capability to assemble novel IPA isoforms and annotate putative protein-coding regions contingent on the identified IPA events. With the augmented annotation, we can quantify the isoform expression levels from the RNA-seq data. A robust correlation of the expression levels of novel assembled IPA isoforms was observed between two HEK293 RNA-seq replicates (Pearson's correlation r = 0.94, P < 0.05), suggesting that they exhibit reproducible expression across different biological replicates (Fig. 2h). In addition, ribosome profiling data of HEK293 cells was utilized to determine the translational efficiency of each identified IPA isoform[38]. In both annotated isoforms and novel IPA isoforms identified by InPACT, the ribosome footprint density was observed to peak around stop codons (Fig. 2i). Further, the ribosome profiling data showcased that the identified intronic terminal exons exhibit a greater transla-tional efficiency than intronic sequences, but a little lower than those already annotated terminal exons (Fig. 2j). For example, the ribosome footprints were found along the intronic terminal exons of the skipped IPA isoform of gene *PIGL* and the composite IPA isoform of gene *CHRNA5*, both of which were experimentally validated by 3'-RACE in HEK293 cells (Fig. 2k, l). Taken together, these findings suggested that the InPACT-identified IPA isoforms are adequately stable to undergo translations.

The interaction between U1 small nuclear ribonucleoprotein (snRNP) and factors involved in cleavage and polyadenylation plays a crucial role in regulating premature 3' end cleavage and poly-adenylation by binding to cryptic intronic polyA sites[39-41]. This process, known as telescripting, is essential for ensuring complete transcription and serves as a general mechanism for controlling transcription elongation[39-41]. Therefore, we employed InPACT to analyze an RNA-seq dataset from HeLa cells treated with antisense morpholino oligonucleotide (AMO) targeting U1, as well as a control group[41] (Supplementary Data 1). Our analysis identified 767 novel IPA events in HeLa cells treated with U1 AMO, while the control group only had 151 events (Supplementary Fig. 7a). Additionally, we examined the dynamic usage of IPA sites in HeLa cells treated with U1 AMO compared to the control group. The observations revealed a significant increase in IPA usage in HeLa cells treated with U1 AMO (Wilcoxon rank sum test, P < 2.2e-16) (Supplementary Fig. 7b). These results are in line with the telescripting activity of U1 snRNP. Overall, these results provided additional compelling evidence supporting the efficacy of InPACT.

## InPACT outperforms current methods on IPA identification and quantification

In order to assess and compare the efficacy of InPACT with other computational methods for IPA analysis based on conventional RNA-seq data, we conducted a comparative analysis using various bench-marks, including experimental 3'-end sequencing data, PacBio long-read sequencing data, and simulated RNA-seq data. We collected several additional RNA-seq datasets with matched benchmarking data from different samples, including the MicroArray/Sequencing Quality Control (MAQC) Universal Human Reference (UHR), MAQC human brain, and human small airway epithelial cells[42] (Supplementary Data 1). It is noteworthy that while several computational methodol-ogies have been developed to investigate APA using RNA-seq data, such as DaPars, QAPA, and APAtrap, they do not extend to IPA analysis. Recently, a method called IPAFinder has been proposed, which is specifically designed for identifying IPA sites from RNA-seq data by considering the changepoint in read coverage[14]. In terms of design principles, InPACT theoretically has the potential to outperform IPA-Finder as it incorporates both genomic sequence and RNA-seq read coverage. Additionally, APAIQ, a recently published method that also leverages the synergistic effect of sequence and read coverage, can be used for identifying IPA sites[28]. Therefore, we primarily focused on comparing InPACT with APAIQ and IPAFinder by evaluating various aspects of performance.

With the curated RNA-seq datasets, we can identify novel IPA sites using InPACT, APAIQ and IPAFinder, respectively. Examination of the polyA signals and nucleotide compositions near IPA sites identified by each tool revealed considerable differences[43]. Specially, both InPACT-identified and APAIQ-identified sites exhibited significant enrichment of the canonical polyA signal (AAUAAA) upstream of the sites, whereas IPAFinder-identified sites did not display enrichment (Fig. 3a). Addi-tionally, the nucleotide profiles surrounding IPA sites identified by InPACT and APAIQ closely resembled those of annotated polyA sites (Supplementary Fig. 8a–p).

In addition, we evaluated and compared the performance of InPACT, APAIQ, and IPAFinder in identifying IPA sites using experi-mental 3'-end sequencing data from the corresponding samples as the ground truth. Different types of experimental 3'-end sequencing data has been utilized, including A-seq, 3P-seq, and PolyA-seq[16,19,44] (Sup-plementary Data 1). For the HEK293 cells, comparison of InPACT-, APAIQ-, and IPAFinder-identified IPA sites with those identified in A-seq and 3P-seq data revealed that over 60% of InPACT-identified IPA sites were located within 50 nt to polyA sites in ground truth, whereas only about 30% of APAIQ-identified IPA sites and 20% of IPAFinder-identified IPA sites were within the same distance. The predicted IPA sites proximal to the ground truth (< 50 nt) were considered as true positive, and it was found that over 80% InPACT-predicted true

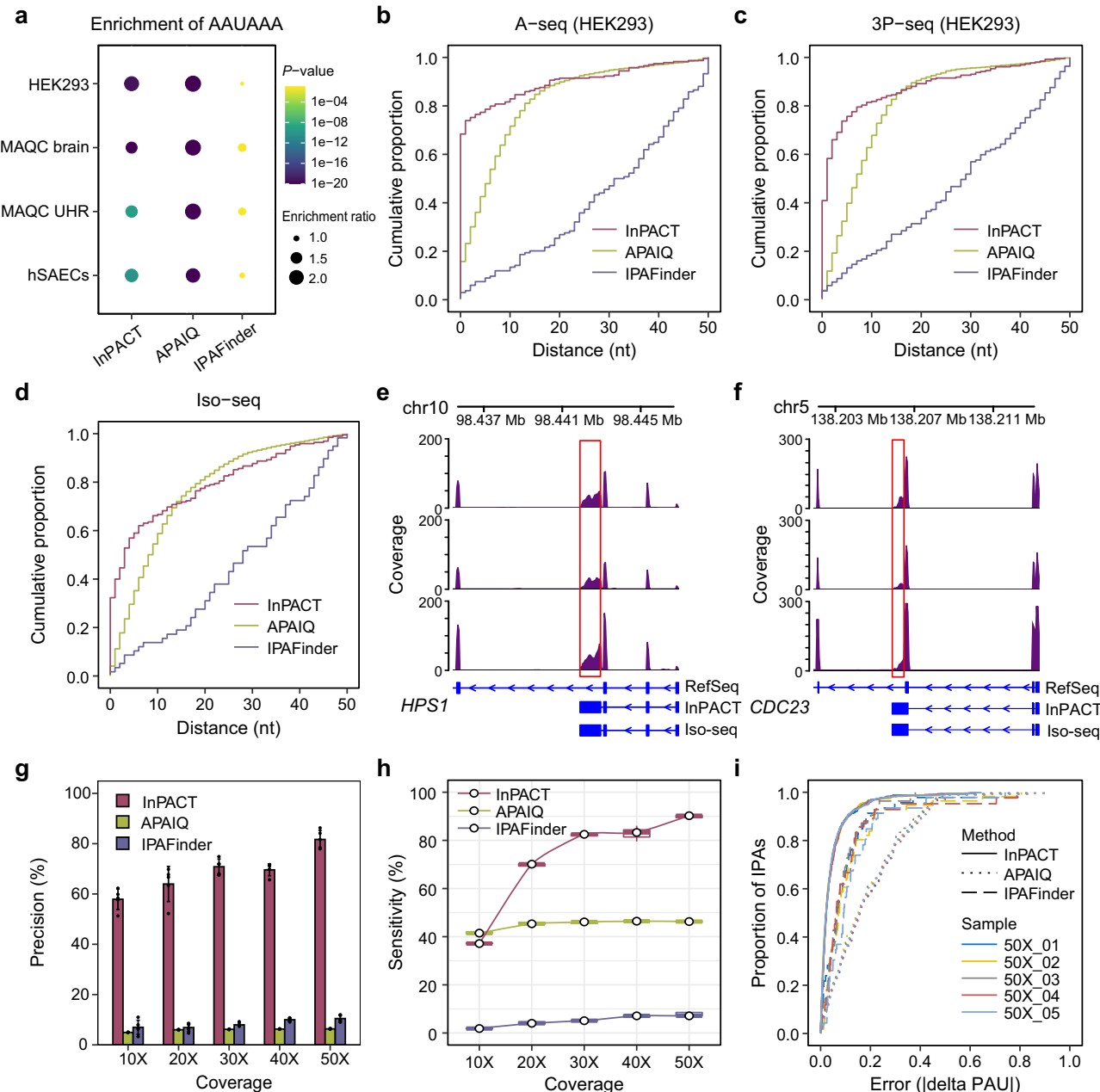

**Fig. 3 | InPACT outperforms other methods on IPA identification and quantification. a** Motif enrichment of canonical polyA signal (AAUAAA) upstream (50 nt) on the IPA sites identified by InPACT, APAIQ and IPAFinder in different RNA-seq datasets by hypergeometric test of MEME Suite (two-sided). **b**, **c** The cumulative distribution curves of the distance between the true positive identified IPA sites and the ground truth, including A-seq (**b**) and 3P-seq (**c**). If an identified IPA site is located within 50 nt from the ground truth, it was regarded as true positive. All the datasets are from HEK293 cells. **d** The cumulative distribution cure of the distance between the true positive identified IPA sites and the ground truth (Iso-seq). This dataset is from human small airway epithelial cells. **e**, **f** Two examples of InPACT-identified IPA isoforms in human small airway epithelial cells. The plots show the read coverage of RNA-seq in the locus for *HPS1* (**e**) and *CDC23* (**f**), with the annotation assembled from long-read Iso-seq and InPACT-identified IPA isoforms.

**g**, **h** The precision (**g**) and sensitivity (**h**) of InPACT, APAIQ and IPAFinder are evaluated for identifying IPA sites using simulated RNA-seq data with varying sequencing coverage levels ranging from 10X to 50X. The identified IPA site located within 50 nt to a predefined polyA site was classified as true positives (TP), while those were not classified as false positives (FP). Replicates were utilized for each coverage level ($n = 5$ random simulations). The precisions are presented as mean values $+/-$ SD (**g**). The sensitivities are presented as box plots. The center lines denote the median values with the boxes are bounded by the 25th and 75th percentiles. The whiskers extend to the maximum and minimum values within 1.5 times the interquartile range (IQR) from each end of the box. **i** The cumulative distribution curve depicts error of the relative usage of IPA sites determined by InPACT, APAIQ and IPAFinder in the simulated RNA-seq data with a sequencing coverage level of 50X.

positive IPA sites located within less than 10 nt to the ground truth, outperforming APAIQ and IPAFinder (Fig. 3b, c). The similar results were also observed in the comparisons of RNA-seq data of MAQC UHR and MAQC human brain (Supplementary Fig. 9). Despite the RNA-seq data and matched 3'-end sequencing data were generated using obviously different sequencing technologies and from different labs,

the results were consistent across different datasets. Collectively, InPACT outperforms APAIQ and IPAFinder in accurately identifying IPA sites from conventional RNA-seq data.

Long-read sequencing methods offer the capability to capture high-quality, full-length transcript sequences, thereby providing reliable isoform information without the requirement for transcript

reconstruction. To evaluate the IPA events identified by InPACT, APAIQ, and IPAFinder, we made use of a dataset that sequenced samples from human small airway epithelial cells in parallel on both short-read and long-read sequencing platforms (Supplementary Data 1). Despite primarily capturing isoforms with high expression, the PacBio platform based single-molecule real-time (SMRT) Iso-seq data validated about 20% of the InPACT-identified novel IPA isoforms, while only approximately 2% of the APAIQ-identified IPA isoforms and 5% of the IPAFinder-identified IPA isoforms were verified. Furthermore, the use of Iso-seq data as ground truth revealed that genomic positions of InPACT-identified IPA sites were more precise than those identified by APAIQ and IPAFinder (Fig. 3d). Specifically, the Iso-seq data of human small airway epithelial cells validated a skipped IPA isoform of gene *HPS1* and a composite IPA isoform of gene *CDC23*, both of which are novel with respect to Refseq annotation (Fig. 3e, f, Supplementary Fig. 10).

Furthermore, we employed simulated RNA-seq data to assess the efficacy of InPACT in detecting IPA sites, with a particular focus on the impact of sequencing depth on the accuracy of identification. The RNA-seq data were simulated with sequencing coverage ranging from 10X to 50X. The precision and sensitivity of InPACT were found to be superior to those of APAIQ and IPAFinder in identifying IPA events, with an increase in sensitivity and precision observed with increasing sequencing depth (Fig. 3g, h). Notably, InPACT successfully identified approximately 90% of IPA sites at a sequencing coverage of 50X (Fig. 3h). Furthermore, InPACT was also evaluated in terms of its ability to identify both skipped and composite IPA sites (Supplementary Fig. 11). Our results demonstrated that InPACT outperforms IPAFinder, regardless of the type of IPA sites. Lastly, we aimed to compare the performance in quantifying IPA events using the simulated RNA-seq data. The ground truth IPA usage can be estimated directly from the simulated RNA-seq data by using the isoform expression level divided by the total expression level of all isoforms from the corresponding gene. The difference between the IPA usage predicted by the tools and the ground truth was used as an error metric to compare the accuracy of InPACT, APAIQ, and IPAFinder in quantifying IPA. The results indicated that InPACT exhibited a lower error rate than both IPAFinder and APAIQ in quantifying IPA events with a sequencing coverage of 50X (Fig. 3i). Additionally, the impact of sequencing depth on quantification accuracy was evaluated by estimating error rates using simulated RNA-seq data with sequencing coverage ranging from 10X to 50X (Fig. 3i, Supplementary Fig. 12). The findings demonstrated that InPACT outperforms both APAIQ and IPAFinder in quantifying IPA events from conventional RNA-seq data.

## Dynamic IPA events in monocytes activation

We next applied InPACT to investigate the transcriptome-wide landscape of IPA in the context of monocytes activation. Specifically, we analyzed RNA-seq from untreated and lipopolysaccharide (LPS)-activated human monocytes, with three replicates per condition[45] (Supplementary Data 1). By utilizing InPACT, we identified a total of 2977 novel IPA sites, with 1105 being skipped and 1872 being composite. We conducted principal component analysis (PCA) on the estimated relative usage profile of these novel polyA sites. The results demonstrated that the biological replicates cluster well with each other, indicating robustness and reliability of our approach (Supplementary Fig. 13a). We focused on the first principal component (PC1) which accounted for the variances and found that the computed PC1 loadings assigned to each IPA event are significantly correlated with the difference between untreated and LPS-activated conditions, providing evidence that the IPA changes are associated with monocyte activation (Supplementary Fig. 13b, c). To further elucidate the underlying patterns of IPA changes during monocyte activation, we conducted a differential transcript usage analysis using DRIMSeq[46], resulting in the identification of 204 significantly differential IPA events (Fig. 4a,

Supplementary Data 3). Additionally, Gene Ontology (GO) enrichment analysis was also conducted, and it revealed that differential IPA events were associated with a variety of biological processes such as neutrophil degranulation, neutrophil activation, defense response and innate immune response (Fig. 4b).

The production of truncated proteins by IPA events may lead to the loss of C-terminal domains or post-translational modification sites, which can affect the functionality of full-length proteins both directly and indirectly. To evaluate the impact of IPA, we conducted an analysis of the percentage of retained coding region (CDR) for each differential IPA isoform in relation to the full-length CDR. The resulting histogram of retained CDR fraction evinced a uniform distribution, albeit with a notable overrepresentation of IPA isoforms that lose all or nearly all the CDR (Fig. 4c). Additionally, we investigated the relationship between IPA and gene expression. A very weak correlation was observed between changes in IPA and changes in gene expression (Spearman's $\rho = -0.079$, $P = 0.0049$) (Fig. 4d). These findings suggested that IPA represents a distinct layer of regulation that is largely independent of gene expression.

As an example, we found that the IPA usage of gene *ARHGAP24* was significantly upregulated in the LPS-activated monocytes (Fig. 4e, f). The *ARHGAP24* gene encodes Rho GTPase-activating protein 24 (ARHGAP24) that possesses a RhoGAP domain responsible for catalyzing the hydrolysis of active guanosine triphosphate (GTP) bound to Rac1, Cdc42, and RhoA, thereby inactivating these regulators[47]. Previous studies have demonstrated that ARHGAP24 can ameliorate inflammatory response through inactivating Rac1[48]. Notably, the truncated protein resulting from the IPA isoform of *ARHGAP24* exhibited a lack of core RhoGAP domain and C-terminal coil structure, as compared to the full-length isoform (Fig. 4g). Consequently, we postulated that the increased usage of IPA may lead to a loss-of-function of *ARHGAP24*, thereby reducing the inhibition of Rac1 activity, which could potentially promote monocyte activation. Furthermore, we conducted 3'-RACE experiments in LPS-activated monocytes to validate four candidate IPA events, including *ARHGAP24*. Consequently, three candidate IPA sites within gene *ARHGAP24*, *RALA*, and *PDCD6IP* were successfully confirmed, whereas the candidate IPA site within the *SDHD* gene failed validation, possibly due to its relatively low expression level (Fig. 4h, Supplementary Fig. 14).

## Application of InPACT to a single-cell RNA-seq dataset

The utilization of single-cell RNA-seq has proven to be crucial in unraveling the heterogeneity and complexities of transcriptomes within individual cells. In order to further demonstrate the utility of InPACT, we have employed InPACT on a human fetal bone marrow (FBM) scRNA-seq dataset consisting of 486 cells[49] (Supplementary Data 1). This dataset encompassed a wide range of cell types, including mast cell ($n = 47$), basophil ($n = 20$), eosinophil ($n = 54$), polymorphonuclear leukocytes (PMN, $n = 65$), myelocyte ($n = 61$), promyelocyte ($n = 44$), monocyte ($n = 32$), B cell ($n = 52$), hematopoietic stem cells (HSC, $n = 32$), plasmacytoid DCs (pDC, $n = 30$), CLEC9A$^+$ DC1 ($n = 34$) and CD1c$^+$ DC2 ($n = 15$) (Fig. 5a). These cells were isolated using fluorescent activated cell sorting (FACS) based on cell-state defining markers and originated from two biologically independent replicates of human FBM.

To comprehensively investigate IPA in human FBM at single-cell resolution[49], we first employed InPACT to identify novel IPA sites. As a result, we discovered a total of 2635 novel IPA events located in 2157 genes, including 599 skipped and 2076 composite events (Supplementary Data 4). The number of novel IPA isoforms expressed in each cell type were depicted in Fig. 5b. For instance, the discovery of *SRP68* IPA and *HMGCL* IPA were supported by a gradual drop-off reads coverage (Fig. 5c, d). Following this, InPACT was further utilized to evaluate the relative usage of all novel IPA events in each individual cell. To elucidate the specific differences in IPA that emerge at the individual

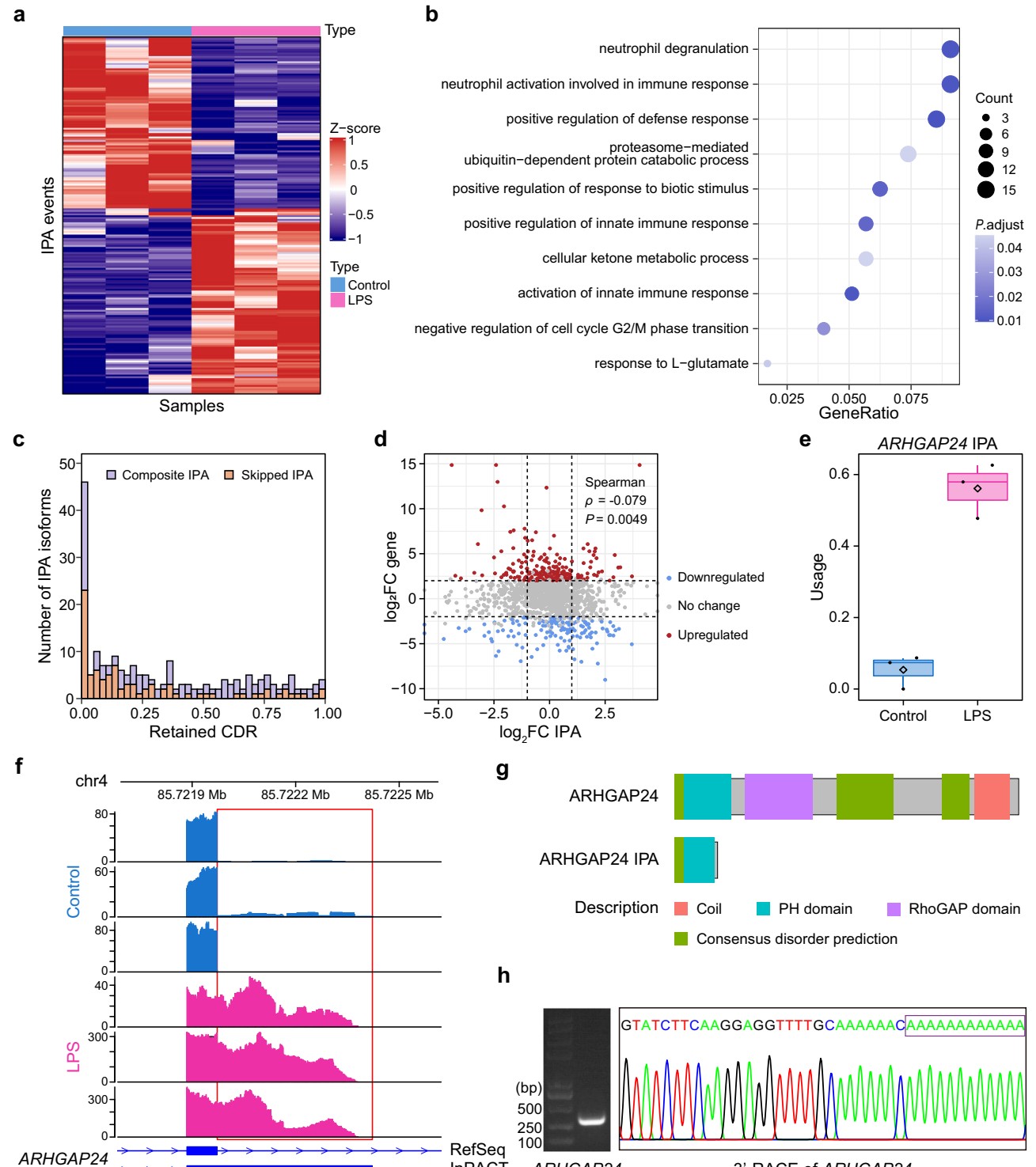

**Fig. 4 | Applying InPACT to identify dynamic IPA events in monocyte activation. a** The heatmap shows 204 significantly differential IPA events between untreated and LPS (lipopolysaccharides)-activated human monocytes ($n = 3$ replicates per condition). **b** The bubble plot depicts GO enrichments (biological process) for genes harboring differential IPA events. The Benjamini-Hochberg adjusted *P*-value estimated by hypergeometric test (two-sided) is shown by blue gradient. **c** The distribution of retained coding region fraction relative to the annotated longest CDR for differential IPA events. The CDR represents coding region. The composite and skipped IPA events are indicated in different colors. **d** The scatter plot depicts the relationship between changes in IPA usage (x-axis) and changes in gene expression (y-axis). Statistically significant differential up-regulated and down-regulated genes are represented by red and blue dots, respectively

($|\log2FC| > 2$, FDR < 0.05, two-sided). The FC represents fold change. Dashed horizontal lines indicate thresholds of gene expression, while dashed vertical lines indicate thresholds of IPA usage. **e**, **f** The comparison of the relative IPA usage of *ARHGAP24* between untreated and LPS-activated human monocytes ($n = 3$ biologically independent samples). The center lines denote the median values with the boxes are bounded by the 25th and 75th percentiles. The whiskers extend to the maximum and minimum values within 1.5 times the interquartile range (IQR) from each end of the box. **g** Schematic representation of full-length and IPA truncated proteins of *ARHGAP24*. Known protein domains are shown as boxes with different colors as indicated. **h** The gel of 3'-Rapid Amplification of cDNA Ends (3'-RACE) experiment for gene *ARHGAP24*, and Sanger sequencing results of the amplified transcripts were depicted. The experiment was repeated $n = 3$ times.

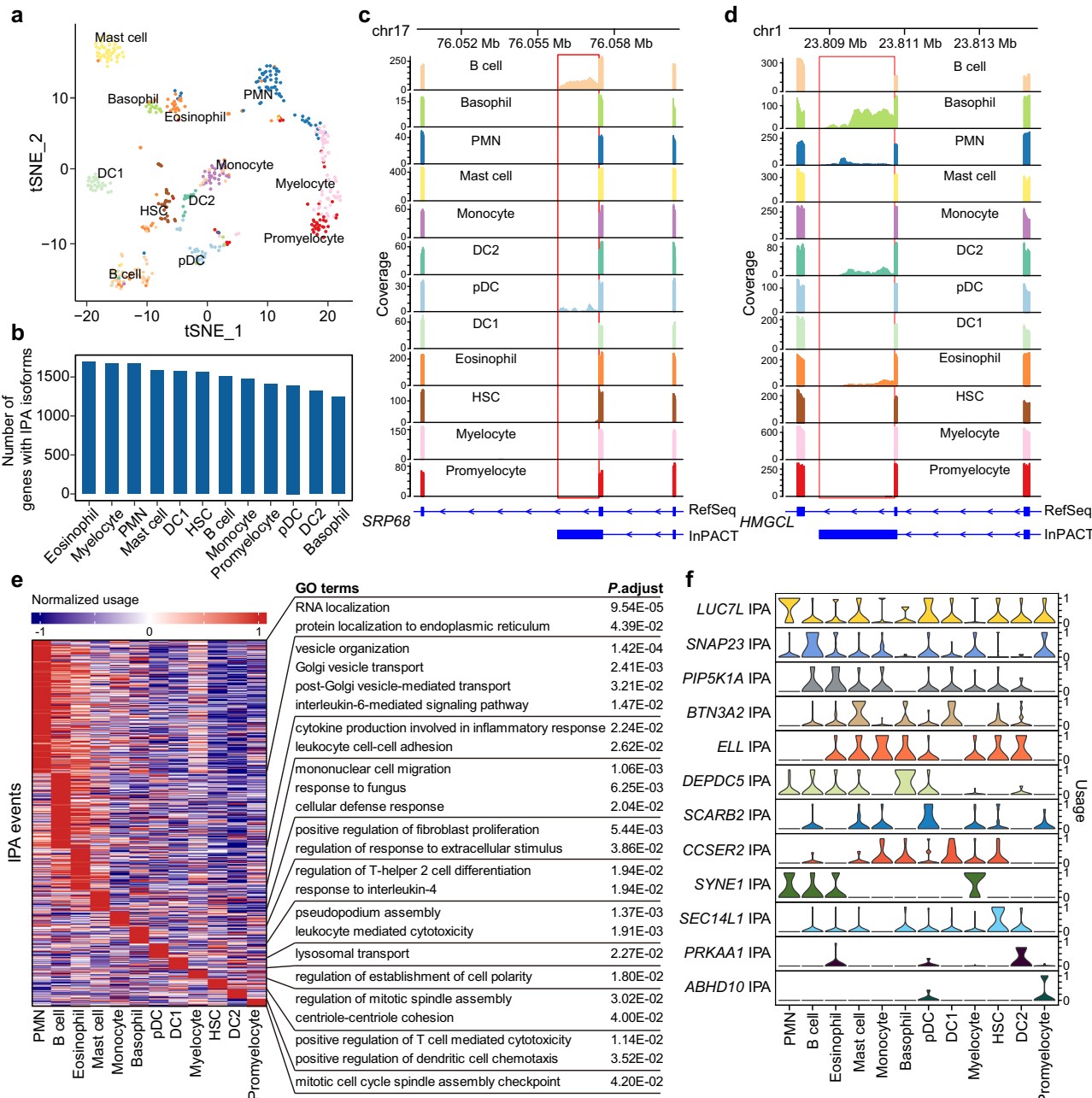

**Fig. 5 | Applying InPACT on a human FBM scRNA-seq dataset identifies cell type-specific IPA events. a** The t-distribute stochastic neighbor embedding (t-SNE) plot of 12 different cell types in human FBM on gene expression level. The HSC represents hematopoietic stem cell, the PMN represents polymorphonuclear leukocytes, the DC represents dendritic cells and the pDC represents plasmacytoid dendritic cells. **b** The bar plot shows the number of genes with novel identified IPA events for each cell type. **c, d** Two examples of InPACT-identified novel IPA events in human FBM scRNA-seq data. The plots show the read coverage in the locus for *SRP68* and *HMGCL*, with the annotation of Refseq isoforms and InPACT-identified novel IPA isoforms. The read coverage is shown for each cell type in different colors. **e** The heatmap depicts cell type-specific IPA events for each cell type. The enriched GO terms (biological process) are listed on the right panel with Benjamini-Hochberg adjusted *P*-values estimated by hypergeometric test (two-sided). **f** The stacked violin plots depict examples of cell type-specific IPA events. The Y-axes represent the relative usage of cell type-specific IPA events.

gene level across various cell types, we undertook a comparison of IPA usage between all cells of one cell type and those of all other cell types sequentially. IPA events displaying significant upregulation in each cell type were designated as cell type-specific events. In the totality of cell types under examination, 533 cell type-specific IPA events were identified (Fig. 5e, Supplementary Data 5).

Furthermore, we performed GO enrichment analysis to evaluate the biological functions associated with genes exhibiting cell type-specific IPA events. The findings revealed that those gene sets are

enriched for biological processes related to the respective cell types. For instance, B cell-specific IPA events were linked to genes involved in antibody secretion and B cell maturation, as evidenced by GO terms such as vesicle organization, Golgi vesicle transport, and IL-6-mediated signaling pathway (Fig. 5e). We also highlighted several specific examples of cell type-specific IPA events, including the *SCARB2* gene, which is known to regulate IFN production of pDC[50]. Interestingly, *SCARB2* was found to utilize IPA sites more frequently in pDC than in other cell types in human FBM (Fig. 5f). Overall, these results suggested

that IPA of genes implicated in distinct biological processes could potentially account for the observed physiological differences among various cell types in the human fetal bone marrow.

## Discussion

In this study, we presented InPACT, a computational approach that incorporates a sequence module and a read module to accurately identify IPA sites from conventional RNA-seq by utilizing both contextual sequence patterns and read alignment information. The efficacy of InPACT was validated through successful detection of IPA sites, which are further confirmed using experimental validations. The authenticity of these IPA sites was confirmed through sequence and expression analyses using multiple lines of evidence, while ribosome profiling data revealed that many of the identified IPA isoforms are in fact translated. Comparative analysis with existing methods, such as APAIQ and IPAFinder, reveals that InPACT outperforms them in precisely identifying IPA sites, as evidenced by comparison to sites captured by 3P-seq, A-seq, polyA-seq, and long-read Iso-seq. Furthermore, comprehensive evaluation using simulated RNA-seq data with varying coverage depths demonstrated InPACT's superior performance, particularly in scenarios of low coverage. Additionally, InPACT exhibited improved accuracy in quantifying IPA events with a lower error rate compared to IPAFinder and APAIQ. Overall, these findings underscore the potential of InPACT as a valuable tool for accurately identifying and quantifying IPA events in conventional RNA-seq data.

Moreover, we delved to the effectiveness of incorporating InPACT into the conventional transcriptome analysis to augment the interpretation of results. By employing InPACT to the RNA-seq data of untreated and LPS-activated human monocytes, the findings revealed several previously unannotated IPA sites. The estimated profile of IPA usage revealed that the IPA difference among different samples aligns with their corresponding biological conditions. The set of genes featuring differential IPA events manifests enrichments for immune response-related genes. Additionally, it is observed that the majority of genes regulated by IPA during monocyte activation do not overlap with differentially expressed genes, which implied that IPA is involved in monocyte activation via an add-on level of regulation. For instance, *ARHGAP24*, a gene causally linked to Rac1 activity, can generate a truncated protein concerning *ARHGAP24* IPA that lacks the catalytic activity of RhoGAP domain. This, in turn, lowers the inhibition of Rac1 activity, which potentially regulates monocyte activation. Undoubtedly, a comprehensive study is needed to clarify the role played by *ARHGAP24* IPA in monocyte activation. These preliminary results, however, shed light on how InPACT can successfully help understand the role of IPA in regulating biological processes.

The advent of scRNA-seq technologies has presented an unprecedented opportunity to explore transcriptomes at the resolution of individual cells, facilitating the elucidation of cell states and diversities. In this study, we have attempted to investigate the landscape of IPA in various cell types of human fetal bone marrow utilizing a recently published scRNA-seq dataset generated by the SMART-seq2 protocol. The application of InPACT to the dataset, with a pooling strategy, effectively uncovered both skipped and composite IPA events. The quantification of IPA at single-cell resolution revealed that several IPA events exhibited a cell type-specific manner. Further, those genes harboring cell type-specific IPA showed enrichment for corresponding biological functions, hinting at the potential roles of IPA in cellular properties and function maintenance. These results highlight the broad applicability of InPACT in characterizing IPA at the single-cell level, thereby allowing for a greater understanding of the context-specific regulation and function of IPA. Notably, scRNA-seq protocols currently in use have been developed based on two primary strategies, full-length and tag-based. InPACT is suitable for full-length data, as it requires full-length coverage of sequencing reads to accurately reconstruct putative intronic terminal exons. However, 3′ tag-based data obtained by enriching RNA 3′ ends can inherently be used for APA analysis. Several bioinformatic methods, such as Sierra, SAPAS and SCAPTURE, have already been developed to address this particular issue[33,51,52]. However, it is worth mentioning that these methods are limited to identifying polyA sites and cannot assemble APA isoforms.

Also of note, InPACT offers a user-friendly approach to characterizing IPA from RNA-seq data. InPACT does not require additional data processing prior to running the program. Moreover, as the sequence module does not rely on RNA-seq data, InPACT has scanned the human reference genome and prepared the set of putative IPA sites of human for users. InPACT also offers the option of constructing a model for a new species, increasing the breadth of its applicability. The read module of InPACT is fully automated and easily accessible, with readily available input RNA-seq alignment files that require no customized processing steps. Additionally, unlike other methods for APA analysis in RNA-seq data, which solely report identified polyA sites and require supplementary manipulation for subsequent analysis, InPACT has integrated the assembly of novel IPA isoforms and the annotation of protein-coding sequences. The resulting augmented GTF file can be readily incorporated into downstream analyses without the need for additional processing steps. This feature enhances the utility of InPACT for IPA analysis and provides a more streamlined and comprehensive approach for users.

Nonetheless, InPACT has several limitations that need to be acknowledged. One such limitation is that InPACT primarily overlooks sample-specific genomic variants, as it utilizes the common reference genome assembly for scanning putative IPA sites in the sequence module. Nevertheless, it is noteworthy that InPACT can adapted to accommodate genome sequences that account for sample-specific variants. Another limitation of InPACT is that its performance is influenced by the quality of the RNA-seq data used. Specially, low RNA integrity and inadequate read coverage may compromise the precision of identifying and quantifying novel IPA events.

Several recent studies have shed light on the prevalence of IPA events across a variety of tissues and cell types [8, 9]. However, many isoforms, which are specific to particular cell types or conditions, have yet to be characterized in the existing genome annotations. A comprehensive examination of IPA is challenging due to the limited availability of specialized 3′-end sequencing technologies. Moreover, using only 3′-end sequencing data may be insufficient for assembling IPA isoforms, which may complicate downstream analyses. Although long-read sequencing can be utilized to detect full-length transcripts, it primarily captures high-abundance transcripts and frequently has lower per-read accuracy than short-read sequencing. With RNA-seq entrenched as the standard method of capturing the transcriptomes, researchers have developed several tools, such as DaPars, Aptardi, and TAPAS, to de novo infer polyA sites in 3′ UTR depending solely on RNA-seq data. Recently, a computational method inspired by DaPars, called IPAFinder, has been proposed for identifying IPA sites by recognizing fluctuations in read coverage. Additionally, a recently published method called APAIQ, which combines DNA sequence and RNA-seq read coverage, can also be used to identify IPA sites. However, APAIQ is not specifically designed for IPA analysis and cannot distinguish between skipped and composite IPA events. Nonetheless, a method to characterize IPA from conventional RNA-seq data is still greatly needed. In this regard, InPACT has been introduced in this study, which is capable of identifying and quantifying IPA with high performance based on conventional RNA-seq data. The broad applicability of InPACT has been demonstrated in inferring the dynamic usage of IPA based on RNA-seq data from different conditions and detecting cell-type-specific IPA events from full-length scRNA-seq data. We envision further applications of InPACT on large-scale transcriptomic datasets such as The Cancer Genome Atlas (TCGA), Genotype-Tissue Expression (GTEx), and Human Cell Atlas (HCA)

could lead to a more comprehensive atlas of IPA, facilitating the understanding of the biological relevance of IPA in human health and diseases.

## Methods

This study complies with all relevant ethical regulations. Ethical approval for all human studies was obtained from the Ethics Committee of Blood Diseases Hospital, Chinese Academy of Medical Sciences, and written informed consent was obtained from the donors.

### InPACT

InPACT is a computational method that could be used to identify IPA sites by combining two modules: the sequence module that scans for all potential polyA sites in genomic regions annotated as introns, and a read module to determine the IPA sites from conventional RNA-seq data (Fig. 1).

**Sequence module.** The sequence module incorporates a convolutional neural network (CNN) with multiple convolutional and pooling layers, and one fully connected hidden layer, which takes genomic sequences in 201 nt windows centered on target sites as input. The sequence is encoded as a binary matrix using one-hot encoding representation. Four nucleotides are converted as followed: A = [1,0,0,0], T = [0,1,0,0], G = [0,0,1,0], C = [0,0,0,1]. The sequences with a length of 201 nt, centered around the polyA sites annotated in Refseq are regarded as positive examples, while a set of 201 nt sequences randomly selected from the intergenic regions, matching the size of the positive examples, as demonstrated in previous studies[33,53].

The CNN model is composed of two pairs of convolutional-pooling layers and one fully connected layer. The sliding window approach is used at the convolutional layer to extract sequence features, which are then fed into the model. The activations $A_f$ of multiple convolutional filters are computed:

$$A_f = \text{ReLU}\left(W_f \cdot S + b_i\right) \quad (1)$$

$$\text{ReLU}(x) = \max(0, x) \quad (2)$$

Here, $W_f$ are the weights of convolutional filters, $S$ is a set of data fed into the model, $b_i$ is the bias, and the rectified linear unit (*ReLU*) is the activation function. To prevent the gradient from disappearing, the activation function is used to increase non-linearity. The activation layers' output sends features to the max-pooling layer, after which the number of parameters could be reduced. The pooling layer aggregates adjacent neurons' activations by extracting the maximum value. Assuming $P$ represents the pooling region and $A_k$ is the activation of the corresponding position, the max pooling $M$ is defined as:

$$M = \max\left(\{A_k | k \in P\}\right) \quad (3)$$

Following the learning of sequence features by two pairs of convolutional-activation-pooling layers, fully connected layers and a dropout function is used. The dropout function is employed to reduce the model's dependence on some neurons to avoid overfitting. The model is fitted on the training set, and hyper-parameters are optimized on the validation set by random sampling. The trained CNN for human has further been evaluated on the polyA sites of GENCODE, PolyA_DB 3 and PolyASite 2.0[30–32]. To ensure unbiased testing, we excluded any overlapping polyA sites used for model training and constructed separate testing sets of polyA sites from GENCODE, PolyA_DB 3 and PolyASite 2.0. For comparison, the performance of DeepPASS, APARENT, and DeepPASTA have also been evaluated on the constructed testing tests.

**Read module.** The read module utilizes the candidate IPA sites from the sequence module as well as the genomic alignments of RNA-seq (BAM format) as input. The read module first constructs putative intronic terminal exons based on the candidate IPA sites and the genomic alignments of RNA-seq data. For each candidate IPA site, InPACT defines a candidate region from the IPA sites to the closest upstream splice site. Candidate regions with enough reads are retained, as determined by featureCount[54]. Then, InPACT can construct putative composite terminal exons or skipped terminal exons based on the read alignments (Supplementary Fig. 2). The skipped terminal exons are constructed when the number of uniquely mapped spliced reads with the 3′ end in the candidate region surpasses a user-defined lower bound (default: five reads). The composite terminal exons are constructed when the number of uniquely mapped unspliced reads that cross the closest upstream splice site surpasses a user-defined lower bound (default: ten reads).

Subsequently, InPACT can determine whether the putative intronic terminal exons exist in the RNA-seq data. This is accomplished by training a machine-learning classifier specific to the analyzed RNA-seq data, leveraging features that characterize the alignments of RNA-seq reads. In order to train the classifier, InPACT first defines three classes of genomic regions based on the Refseq annotation and the corresponding RNA-seq data. These classes include terminal exons (annotated unique last exons that have at least five splice-in reads), internal exons (annotated unique exons located between the first and last exons that have at least five splice-in reads) and background regions (annotated unique last exons that have less than five splice-in reads). The training and testing sets are created by randomly splitting this collection of these genomic regions in an 80:20 ratio. InPACT computes various features from the alignments of RNA-seq data to characterize each region. These features enable discrimination between true terminal exons, internal exons, and background regions. The features encompass relative region length, normalized region expression, coefficients of variation, entropy efficiency and others that predominantly characterize the spliced and unspliced reads across the 5′ end and 3′ end (Supplementary Fig. 3). To account for the differences between skipped and composite terminal exons, the classifiers with different feature sets are trained for putative skipped and composite terminal exons, respectively. To increase the stability of the model, InPACT trains classifiers on ten randomized subsamples of the training set using random forest, and then ensemble the classifiers. With the testing set, a series of metrics for the evaluation could be measured, including true negative rate, true positive rate, Accuracy, Precision, F1 score and Matthew's correlation coefficient.

$$\text{TNR} = \frac{\text{TN}}{\text{TN} + \text{FP}} \quad (4)$$

$$\text{TPR} = \frac{\text{TP}}{\text{TP} + \text{FN}} \quad (5)$$

$$\text{ACC} = \frac{\text{TP} + \text{TN}}{\text{TP} + \text{TN} + \text{FP} + \text{FN}} \quad (6)$$

$$\text{Precision} = \frac{\text{TP}}{\text{TP} + \text{FP}} \quad (7)$$

$$\text{F1 score} = \frac{2 \times \text{TP}}{2 \times \text{TP} + \text{FP} + \text{FN}} \quad (8)$$

$$\text{MCC} = \frac{\text{TP} \times \text{TN} - \text{FP} \times \text{FN}}{\sqrt{(\text{TP} + \text{FP})(\text{TP} + \text{FN})(\text{TN} + \text{FP})(\text{TN} + \text{FN})}} \quad (9)$$

TP, TN, FP, and FN denote the number of true positives, true negatives, false positives, and false negatives, respectively. With the trained classifiers, InPACT can determine the genuine intronic terminal exons from the candidate regions, thereby identifying the IPA sites.

**Quantification of IPA events.** Through the sequence module and read module, InPACT can accurately identify novel IPA sites and intronic terminal exons. In the final step, InPACT can assemble novel IPA isoforms based on the reference annotation and search for the first in-frame stop codon in the isoform. The detailed annotation for each IPA isoform is outputted in General Transfer Format (GTF) by InPACT. Salmon, a fast and GC bias-aware quantification procedure using dual-phase inference, is implemented to compute the transcript-level abundance of all isoforms[55]. To quantify the relative usage of an IPA isoform, InPACT calculates the relative expression of an IPA isoform by comparing it to the total expression level of all isoforms within a gene. This metric, referred as IPA usage, allows for the assessment of the relative usage of IPA isoforms:

$$\text{IPA usage} = \frac{x_{ig}}{\sum_j x_{ig}} \qquad (10)$$

Where $g$ is a give gene, $x_{ig}$ is the expression level of isoform $i$ within gene $g$, measured in transcripts per million (TPM).

### Conservation analysis
PhyloP scores could be used to assess evolutionary conservation, with positive and negative scores respectively indicating that predicted sites are conserved and fast-evolving[56]. To establish a suitable negative control, we initially randomly selected a set of genomic sites within annotated intronic regions, ensuring that the group's size matched that of the analyzed IPA sites. Subsequently, we obtained the PhyloP 30-way track for the human genome from the UCSC Genome Browser and proceeded to calculate the conservation level for each site by averaging the scores of the 20 nt upstream and downstream of the site, utilizing the bigWigAverageOverBed tool[57].

### Analysis of translation using ribosome profiling data
The ribosome profiling data of HEK293 cells were used to determine whether the InPACT-identified novel IPA isoforms are actively translated[38]. We first removed those reads mapped to rRNA sequences using Bowtie[58], and then aligned the reads to the human genome (GRCh38) using HISAT2[59]. The genome index was built based on the Refseq annotation. Reads flagged as secondary alignment were filtered out using SAMtools[60], ensuring one genomic position per aligned read. RiboWave, a ribosome profiling data processing tool that denoise the original signal by wavelet transforms, was then used to further denoise the ribosome profiling data[61]. Only the identified IPA isoforms with an in-frame stop codon were considered for translation analysis. We counted mapped ribosome-protected reads around the stop codons in both annotated isoforms and InPACT-identified novel IPA isoforms and plot the ribosome footprint density around stop codons, respectively.

### Benchmarking InPACT using A-seq, 3P-seq, polyA-seq, and long-read Iso-seq data
To benchmark InPACT in terms of identifying IPA sites, multiple datasets of cell lines and tissue samples were utilized as the ground truth for comparison. These datasets were generated through different 3' end sequencing protocols, namely A-seq, 3P-seq, and polyA-seq. Detailed information regarding each dataset, including the accession number, can be found in Supplementary Data 1. The genomic coordinates of polyA sites identified in A-seq, 3P-seq, and polyA-seq were obtained from the Gene Expression Omnibus (GEO) and subsequently converted to the GRCh38 genome assembly using the liftOver tool[57].

Subsequently, the IPA sites identified by InPACT, APAIQ, and IPAFinder from RNA-seq data were compared with those captured in the 3' end sequencing data. Each of the A-seq, 3P-seq, and polyA-seq datasets served as the respective ground truth. The distances between the identified polyA sites and the closest reference were determined using BEDTools *closest*[62].

We further utilized a dataset sequencing human small airway epithelial cells in parallel using both conventional short-read RNA-seq and long-read Iso-seq protocols to benchmark the performance of InPACT. This dataset was downloaded from GEO under accession number GSE167486 (Supplementary Data 1). For the short-read data, IPA sites can be identified using InPACT and IPAFinder, respectively. For the long-read data, polyA sites can be directly extracted from the released isoform annotation file assembled from the Iso-seq data. Then, we compared the IPA sites identified using InPACT and IPA-Finder with the polyA sites detected from Iso-seq using BEDTools *closest*[62].

### Benchmarking InPACT for IPA analysis using simulated RNA-seq data
The precision of identifying IPA sites can be influenced by the sequencing depth of RNA-seq data. In order to further compare InPACT with APAIQ and IPAFinder, we simulated several RNA-seq data with different sequencing coverage levels using the R package Polyester[63]. The RNA-seq data were simulated with varying sequencing coverage levels ranging from 10× to 50× with a 10× increment, and five replicates were generated for each coverage level to ensure accuracy and reliability of the results. Subsequently, InPACT, APAIQ and IPAFinder were employed to predict IPA sites for each simulated RNA-seq data. The IPA sites predicted within 50 nt of a predefined polyA site were classified as true positives (TP), while those outside this range were classified as false positives (FP). Sensitivity and precision were calculated for each coverage level using the formulas: Sensitivity = TP / Predefined and Precision = TP / (TP + FP).

For benchmarking IPA quantification, we first utilized InPACT, APAIQ and IPAFinder to compute the relative usage of each identified IPA site based on the simulated RNA-seq data. The determined relative usage of predefined polyA sites were considered as the ground truth. We then utilized a metric inspired by a previous study, referred to as error, to assess the concordance between estimated IPA usage and the ground truth[64]. For each IPA site, error was defined as the absolute difference between the estimated relative usage and the ground truth ($|\Delta\text{PAU}|$). The cumulative distributions of error across all identified IPA sites were compared between InPACT and IPAFinder.

### IPA analysis of monocytes' RNA-seq data
The RNA-seq data of six samples from human monocytes (three control samples and three samples stimulated with 100 ng/ml LPS for 6 h) were downloaded from GEO under accession GSE118165[45]. The raw RNA-seq data were aligned to human reference genome (GRCh38) using HISAT2[59]. Then we applied InPACT to identify novel IPA sites and assemble novel IPA isoforms from these RNA-seq data. After computing the expression level of these novel IPA isoforms, we applied DRIMSeq to conduct differential transcript usage analysis between control and LPS stimulated group[65]. DRIMSeq employs a statistical framework based on the Dirichlet-multinomial distribution, which allows for the identification of changes in isoform usage between conditions[46]. As a result, we could define the significantly differentially expressed IPA isoforms by the cutoff of $|\log_2\text{fold\_change}| > 1$ and FDR < 0.05, thereby identifying the differentially used IPA sites. Gene Ontology enrichment analyses were performed on the genes with differentially used IPA sites using ClusterProfiler (version 3.18.1)[66]. The significant enriched GO terms of biological processes were defined by FDR < 0.05. The InterPro, an integrated resource for protein families, domains and functional sites, was used to predict protein domains[67].

## IPA analysis of human FBM scRNA-seq data

The scRNA-seq of human FBM were downloaded from EMBL-EBI ArrayExpress under accession E-MTAB-9801[49]. The scRNA-seq reads were mapped to the human reference genome (GRCh38) using HISAT2 for each individual cell, followed by the removal of PCR duplicates using samtools *rmdup*[59,60]. Subsequently, the aligned reads were pooled together, and InPACT was applied to identify novel IPA sites and reconstruct novel IPA events. Using the augmented annotation file by combining novel IPA events, we could estimate the relative usage of each IPA event in each individual cell. To determine cell type-specific IPA events, low-abundance genes with TPM < 1 were filtered out. Then, we conducted a comparison of IPA usage between cells of the same cell type and those of other cell types sequentially to identify the cell type-specific IPA events. The significance of GO term enrichment was determined using the hypergeometric test.

## Experimental validation of candidate IPA sites

The HEK293T cells were obtained from the National infrastructure of Cell Line Resource (1101HUM-PUMC000010, China) and cultured in cultured in 10% FBS DMEM medium (MA0212, MeilunBio, China). Human peripheral blood samples were collected from healthy donors and treated with EDTA anticoagulant to prevent clotting. The blood sample was mixed with an equal volume of PBS. Diluted blood was added to the upper layer of the Ficoll density gradient solution (17144002, Cytiva, USA) and then centrifuged at 20 °C, 400 × g for 30 min. White PBMC layer was collected and washed with cold PBS for two times. Human monocytes were isolated from the PBMC and cultured in 10% FBS RPMI-1640 medium (11875119, Gibco, USA). Subsequently, monocytes were stimulated with 100 ng/ml LPS (L2630, Sigma-Aldrich, USA) for 6 h. Total RNA from both HEK293T cells and human monocytes were isolated using TRIzol (P118-05, GenStar, China) according to the manufacturer's instructions. 3'-RACE was performed using 3'-Full RACE Core Set with PrimeScript™ RTase (6106, TaKaRaBio Technology, China) according to the manufacturer's instructions. Briefly, 1 μg of total RNA from HEK293T cells or monocytes was reversely transcribed into cDNA using 3'-RACE adaptor. Then the cDNA of a specific gene was amplified using 3' RACE outer primer and gene specific outer primer, followed by a nested PCR using 3' RACE inner primer and gene specific inner primer. The PCR products were ligated into pCDH-CMV-MCS-EF1-Puro or pUCm-T vector and sequenced. The sequences of all primers were described in Supplementary Data 6 and 7.

## Reporting summary

Further information on research design is available in the Nature Portfolio Reporting Summary linked to this article.

## Data availability

A detailed description of datasets used in this study is provided in Supplementary Data 1. Specially, the RNA-seq, ribo-seq, A-seq, and 3P-seq of HEK293 cells can be downloaded from GEO under accession number GSE56010, GSE73136, GSE37037, GSE52527, respectively. The RNA-seq dataset of HeLa cells treated with control and U1 AMO can be downloaded from GEO under accession number GSE193200. The RNA-seq and polyA-seq datasets of MAQC UHR and human brain can be downloaded from GEO under accession number GSE49712 and GSE30198, respectively. The RNA-seq and matched PacBio SMRT Iso-seq dataset of human small airway epithelial cells can be downloaded from GEO under accession number GSE167486. The RNA-seq dataset of untreated and LPS-activated human monocytes can be downloaded from GEO under accession number GSE118165. The scRNA-seq data of human fetal bone marrow can downloaded from EMBL-EBI ArrayExpress under accession number E-MTAB-9801. The Sanger sequencing data of 3'-RACE products have been deposited in Zenodo [https://doi.org/10.5281/zenodo.10801168]. All data generated during this study

are included in this published article and its supplementary information files. Source data are provided with this paper.

## Code availability

InPACT is implemented as an open-source tool that can be obtained from GitHub repository (https://github.com/YY-TMU/InPACT) and also from Zenodo[68].

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

## Acknowledgements

This work was supported by the National Natural Science Foundation of China (Grant No. 32100534 to Y.Y., Grant No. 32200514 to J.Y., Grant No. 32270861 and 32070647 to C.X., Grant No. 32270970 to L.L. Grant No. 32200547 to Y.W.). We thank all members in the group for their assistance and constructive suggestions. We also gratefully acknowledge the technical support by the High-performance Computing Platform of Tianjin Medical University.

## Author contributions

Y.Y., J.Y., and C.X. conceived and designed the study. Y.Y. supervised the study and data analysis. X.L. developed the method, wrote the original code, and analyzed the data. H.C. performed the experiments with help from J.Z. and L.L. Z.L., W.J. and X.Y. helped the data analysis. Y.W. helped the method development. Y.Y., J.Y., and X.L. wrote the manuscript with input from all authors. All authors approved the final version submitted.

## Competing interests

The authors declare no competing interests.
