## [Peer Review File · Nature Communications]

InPACT: A computational method for accurate characterization of intronic polyadenylation from RNA sequencing dataReviewer #1 (Remarks to the Author):

Liu et al. introduced a novel method called InPACT. This manuscript aims to detect previously unannotated IPA sites utilizing RNA-seq datasets. There are several issues in the methodology/data of this manuscript.

- The authors have not provided any explicit details about their training data except that they have used annotated polyA sites. There is no mention of what comprised their negative examples for training.
- Using the annotated polyA sites to train and then using RefSeq, GENCODE, PolyA_DB3 and PolyASite 2.0 for testing the performance of the model seems inappropriate as these datasets have a very high degree of overlap with the training data.
- Like the sequence module, the training data for the read module has not been explained at all.
- Testing the model on two RNA-seq HEK293 datasets seems insufficient to assess the performance of the tool. There is no information on where this dataset is coming from. There is so much RNA-seq data available and testing on just cell line data is insufficient.
- InPACT should be compared to ground truth, which is polyA-seq, 3'-seq etc in this case and not just IPAFinder.
- Although the authors claim that InPACT can be used for characterization and quantification of IPA sites. There was no information on how quantification was done.
- Throughout the manuscript, there are lot of details and information missing making it very difficult to follow through the different steps. For example, authors have reported p value <0.05 in line 179 without any mention of what statistical test they used etc. many more examples of missing information has been mentioned above.

Reviewer #2 (Remarks to the Author):

In this manuscript, Liu and colleagues describe a new computational approach for identifying and quantifying the use of intronic polyA (IPA) sites. This method, termed InPACT, uses a variety of features, both learned from a reference genome sequence and observed in RNAseq data, to identify unannotated IPA sites. After demonstrating the power of the approach, they then use it to identify IPA sites that are differentially regulated upon monocyte activation with LPS. Finally, they apply their approach to a single-cell RNAseq dataset from fetal bone marrow, demonstrating that many of these new IPA sites are differentially used across cell types. Overall, I found that the manuscript made a clear case for why this technique is superior to what is currently available. I believe that the method will be a valuable addition to the field. I have a few suggestions that may improve the manuscript.

MAJOR COMMENTS

1. In figure 2E, the authors compare conservation scores of sequences surrounding previously annotated and newly identified IPA sites. This is fine, but it's hard to really interpret this result without a negative control sequence set. Yes, newly identified IPA sites are less conserved than previously annotated ones, which might be expected. A more interesting and interpretable question, though, would be to ask whether the newly identified sites are more conserved than expected (i.e. to other intronic sequences that may act as suitable negative controls). The new IPA sites may be expected to be more conserved than these negative control sites, and if so, this would be additional evidence of their existence and function.

2. In figure 4D, the authors attempt to show that changes in IPA usage do not have a noticeable effect on gene (RNA) expression as there is little overlap between the genes that display differential IPA usage and those that are differentially expressed. This analysis could be improved. For example, if you just look at genes that are differentially expressed, you are combining those that increase with expression and those that decrease in expression. If the connection between IPA usage and gene expression is directionally correlated, then you might expect a result where, for example, usage of an upstream IPA site tends to increase gene expression while usage of a downstream site decreases gene expression. This exact result has actually been observed before (Goering et al, BMC Genomics 2021, figure S4B and S4D).

Both thresholding on a significance cutoff for differential expression and IPA usage and lumping all gene expression changes together may obscure this result.

MINOR COMMENTS

1. In a few places, the authors fail to cite work that has demonstrated clear function for IPA. This can perhaps partially be attributed to the fact that earlier literature on the subject sometimes referred to these isoforms as ALE (alternative last exons) rather than IPA. However, the two terms refer to the same thing. Although IPA isoforms differ in their C-terminal coding regions, they also differ in 3' UTR content (which is not discussed in the manuscript). One consequence of this difference in 3' UTR content is that transcripts that use different IPA (ALE) sites are often differentially localized within cells (Taliaferro et al, Mol Cell 2016). The reason I bring this up is that the introduction reads like the function of IPA is unknown.

2. The authors state that currently available APA analysis tools are not well-suited for IPA quantification. This is true for the tools listed, but at least one that was not listed (LABRAT, Goering et al, BMC Genomics 2021), can quantify differential IPA (ALE) isoform usage. Related to Major Comment 2, IPA usage again was shown to be connected to transcript abundance. A major advantage of InPACT over LABRAT is InPACT's ability to identify new IPA isoforms. This is something LABRAT cannot do.

A much older software package aimed at quantifying alternative splicing, MISO, also has the ability to quantify ALE usage, but again cannot identify new isoforms.

3. This is just a suggestion, and not something that should be required for publication, but given InPACT's ability to identify new IPA isoforms, it could be interesting to look at what InPACT says in datasets in which U1 snRNP function has been inhibited. There is quite a bit of literature about U1's ability to block the usage of cryptic, upstream, often intronic polyA sites (so-called "telescripting", for review see PMID 30709878). If for example, InPACT detected many new IPA isoforms in samples where U1 was inhibited, this would be even further evidence of the high performance of InPACT and would really highlight its ability to find new IPA sites.

Reviewer #3 (Remarks to the Author):

Liu et al present a computational method for accurate characterization of intronic polyadenylation from standard RNA-seq data. While they showed that InPACT is a powerful tool, they ignore the most relevant tool called APAIQ that recently published in Genome Research for the direct comparison. This emphasizes a fact that InPACT is not the first method utilizing the synergistic effect of RNA-seq read coverage and DNA sequence on IPA site identification and should be mentioned. In addition, a comprehensive comparison between InPACT and APAIQ should be carried out before the consideration of publication in Nature Communications. Therefore, I would suggest 'major revision' for this manuscript.

Major comments:

1) As we know, there exists synergistic effect of RNA-seq read coverage and DNA sequence on PAS identification. IPAFinder performed de novo IPA identification and quantification from standard RNA-seq data based on "change point" model, which also have been used to analyse tandem APA events by DaPars or PAQR. Given that IPAFinder only considered RNA-seq read coverage information, thus IPAFinder is sensitive to coverage bias of mRNA 3' end, which may influence the identification of accurate position of IPA sites. InPACT developed in this manuscript considered more features than IPAFinder, which included both RNA-seq read coverage information and genomic information associated with mRNA 3' end processing. Thus, InPACT deservedly better than IPAFinder. I think that authors should refer to this point in the main text.

2) Authors didn't mention statistics model about detecting significantly differentially used IPA sites in the method section. I think that the appropriate statistics model is very important for analyzing small number of samples (5 controls vs 5 cases).

3) Authors didn't mention APAIQ method (published by a recent Genome Research paper (PMID: 37117035)), which could be used to identify both UTR APA and IPA sites. Most importantly, APAIQ also integrates RNA-seq read coverage information with genomic sequence through CNN. Thus, InPACT developed here was not the first method utilizing the synergistic effect of RNA-seq read coverage and DNA sequence on PAS identification. I think authors should compare their InPACT with APAIQ, which is more appropriate than comparison with IPAFinder. The direct comparison between InPACT and APAIQ should be put into the central position of the result section. Comparison to IPAFinder could be moved to the Discussion section or Supplementary Materials.

4) The authors didn't mention deep learning models APARENT (PMID: 31178116) and DeepPASS (PMID: 34376223). APARENT model could predict polyA site strength based on genomic sequence. DeepPASS model could precisely identify true polyA site based on genomic sequence with high sensitivity and accuracy, which further be used to analyze differential transcript expression at single-cell resolution. Given that the result about that applying sequence module of InPACT on PolyASite 2.0 prediction obtain an AUROC of 0.794, I have concerns about the performance of CNN model of InPACT. I think performing the comparison between InPACT with APARENT and DeepPASS is necessary.

5) Authors have performed some analyses about applying InPACT on scRNA-seq data. Nowadays, a large amount of scRNA-seq data was accumulated in the public databases. I wonder whether InPACT is expandable on the other type (3' tag but not full-length) of widely used scRNA-seq data (such as 10x Genomics), which will extend the application range of InPACT and attract more interest of potential users.

6) The authors should perform experimental validations for at least ten genes in diverse cell types or biological processes such as KEK293 cells (Figure 2), human small airway epithelial cells (Figure 3), monocytes activation (Figure 4), and human fetal bone marrow cells (Figure 5). For a methodology paper, ten genes are reasonable number for the experimental validation. In Fig 2A, 2B, 2I and 2J, RT-PCR and Sanger sequencing should be performed to demonstrate the reliability of IPA sites in ZNF771, TERF2, PIGL and CHRNA5 genes. In Fig. 3E and 3F, RT-PCR and Sanger sequencing should be performed to demonstrate the reliability of IPA sites in HPS1 and CDC23 genes. In Fig. 4F, RT-PCR and Sanger sequencing should be performed to demonstrate the reliability of IPA sites in ARHGAP24 and more other genes. In Fig. 5C and 5D, RT-PCR and Sanger sequencing should be performed to demonstrate the cell-type specificity of IPA sites in SRP68 and HMGCL genes. Results regarding the B cell-specific IPA site in SRP68 and Basophil, DC2, Eosinophil-specific IPA sites in HMGCL should be shown.

Point-by-point address of the comments raised by the reviewers for manuscript NCOMMS-23-25772

We express our sincere gratitude to you and the reviewers for diligently evaluating our manuscript and offering valuable feedback. We are delighted to report that we have effectively incorporated all the comments made by the reviewers. Their insightful comments, along with constructive suggestions, have significantly enhanced the quality of our manuscript. Substantial revisions have been implemented in the manuscript, specifically addressing the concerns raised by the reviewers. Please see below for a point-by-point response to the reviewers' comments.

Reviewer #1 (Remarks to the Author):

Liu et al. introduced a novel method called InPACT. This manuscript aims to detect previously unannotated IPA sites utilizing RNA-seq datasets. There are several issues in the methodology/data of this manuscript.

RESPONSE: We thank the reviewer for the valuable comments and insightful suggestions regarding our manuscript, which have proven to be invaluable in enhancing the quality of our study. We have diligently considered all the comments provided and have implemented significant revisions throughout the manuscript. The major changes are highlighted in red for you to access easily.

1. The authors have not provided any explicit details about their training data except that they have used annotated polyA sites. There is no mention of what comprised their negative examples for training.

RESPONSE: Thank you for the insightful comments. We have taken note of the comment regarding the lack of explicitness in the description of the training data for the *Sequence module*. In response, we have made revisions to the manuscript to provide a comprehensive account of both the positive and negative examples used for model training (Page 18 Line 520-523).

For the training of the *Sequence module* of InPACT, we considered the polyA sites annotated in Refseq as the positive sets. Specifically, we utilized genomic sequences with a length of 201 nt, centered around the polyA sites annotated in Refseq, as positive examples. To ensure a balanced training and testing process, it was necessary to incorporate negative examples for the model as well. To achieve this, we randomly extracted a set of 201 nt sequences from the intergenic regions, matching the size of the positive examples, as previously demonstrated in a similar investigation [1].

To facilitate the input of sequence examples into the CNN model, we employed the one-hot encoding representation (A = [1,0,0,0], T = [0,1,0,0], G = [0,0,1,0], C = [0,0,0,1]), where each nucleotide was represented by a binary matrix of dimensionality 4×201 . This encoding scheme allowed for efficient processing and analysis within the model.

2. Using the annotated polyA sites to train and then using RefSeq, GENCODE, PolyA_DB3 and PolyASite 2.0 for testing the performance of the model seems inappropriate as these datasets have a very high degree of overlap with the training data.

RESPONSE: We appreciate the valuable feedback provided by the reviewer. To train the Convolutional Neural Network (CNN) model in the *Sequence module*, the annotated polyA sites from Refseq were utilized. We acknowledge the reviewer's concern regarding potential overlap between polyA sites from GENCODE, PolyA_DB 3 and PolyASite 2.0, and the polyA sites used for training the model from Refseq [2-4]. Consequently, we have taken the necessary steps to address the issue by excluding the overlapping polyA sites from GENCODE, PolyA_DB 3, and PolyASite 2.0, respectively, thereby constructing a separate set of testing polyA sites. Subsequently, we evaluated the performance of the CNN model in the *Sequence module* using the aforementioned testing polyA sites from GENCODE, PolyA_DB 3, and PolyASite 2.0, respectively. Our results demonstrate that the CNN model effectively predicts the testing polyA sites, achieving an AUROC of 0.954 for GENCODE, 0.920 for PolyA_DB 3, and 0.794 for PolyASite 2.0, with no significant changes or only marginal decreases observed (**Page 6 Line 138-144, Supplementary Figure S1B-D**).

In accordance with the recommendation of Reviewer #3, we conducted a comparative analysis of the *Sequence module* of InPACT with several recently published deep learning

models. Specifically, we selected three models, namely DeepPASS [1], APARENT [5], and DeepPASTA [6], for the purpose of comparison. DeepPASS was trained on a rigorous collection of polyA sites that are annotated in all three resources, including PolyA_DB 3, PolyASite 2.0, and PolyA-seq paper, or in the GENCODE annotation [1, 7]. APARENT was trained using a massively parallel reporter assay, encompassing over 3 million synthetic APA reporter constructs [5]. Lastly, DeepPASTA was trained on the polyA sites annotated in the PolyA-seq paper [6, 7]. Upon comparing InPACT with DeepPASS, APARENT, and DeepPASTA, we observed that InPACT exhibits a higher or comparable performance, indicating that the *Sequence module* of InPACT has the ability to predict polyA sites with a high level of confidence (**Page 6 Line 144-151, Supplementary Figure S1B-D**).

3. Like the sequence module, the training data for the read module has not been explained at all.

RESPONSE: We appreciate the valuable comments provided. It has come to our attention that the previous version of the manuscript lacked explicit information regarding the training data for the Read module. In response, we have made revisions to the manuscript to provide a detailed explanation of the training data for the *Read module* (**Page 20 Line 562-576**).

The *Read module* plays a crucial role in identifying genuine intronic polyA sites expressed in the RNA-seq data under analysis. This is accomplished by training a model specific to each sample, leveraging features that characterize the alignments of RNA-seq reads. These features enable the discrimination between true terminal exons, internal exons, and background regions. To train the sample-specific model using the input RNA-seq data, three types of genomic regions are constructed: terminal exons, internal exons, and background regions. The definitions of these regions are based on the Refseq annotation and the corresponding RNA-seq data. Terminal exons refer to the last exons of mRNAs. However, terminal exons that either overlap with other exons or fail to meet the minimum requirement for splice-in-boundary reads (default: five reads) are excluded from the analysis. Internal exons are situated between the first and last exons of mRNAs. For inclusion in the analysis, internal exons must not overlap with other exons and must possess the minimum number of splice-in-boundary reads (default: five reads). Background regions, like terminal exons, are

the last exons of mRNAs. However, they do not overlap with other exons and do not meet the minimum requirement for splice-in-boundary reads (default: five reads).

By incorporating these three types of genomic regions, the training and testing sets are created by randomly splitting this collection of these genomic regions in an 80:20 ratio. InPACT computes various features from the alignments of RNA-seq data to characterize each region. These features enable discrimination between true terminal exons, internal exons, and background regions. The features encompass relative region length, normalized region expression, coefficients of variation, entropy efficiency and others that predominantly characterize the spliced and unspliced reads across the 5' end and 3' end (Supplementary Figure S3).

4. Testing the model on two RNA-seq HEK293 datasets seems insufficient to assess the performance of the tool. There is no information on where this dataset is coming from. There is so much RNA-seq data available and testing on just cell line data is insufficient.

RESPONSE: We express our gratitude to the reviewer for providing valuable feedback. The two RNA-seq datasets pertaining to HEK293 cells were obtained from a previously published study and can be accessed through the Gene Expression Omnibus (GEO) under accession number GSE56010 [8]. We acknowledge that the information regarding these RNA-seq datasets of HEK293 cells was previously only presented in the “Data availability” section with reference paper. In response, we have made revisions to the manuscript to incorporate detailed information about the HEK293 RNA-seq data. In conjunction with the “Data availability” section, we have included a citation to the reference paper that provides the dataset within the Results section (**Page 7 Line 172-174**), while incorporating the accession details into **Supplementary Table S1**

Furthermore, we concur with the reviewer’s comment that many RNA-seq datasets are available, and it is needed to utilize different RNA-seq datasets to assess the performance of InPACT. However, for the purpose of benchmarking, we specifically selected RNA-seq data that had matched experimental 3' end sequencing data as the ground truth. In the previous version, RNA-seq data from both cell lines and tissue samples, including HEK293 cell lines (GSE56010) and human small airway epithelial cells (GSE167486), were incorporated to evaluate the performance of InPACT (**Page 11 Line 287-300**). To enhance the robustness of

our results, we have now included several additional RNA-seq datasets from various samples, along with their paired 3' end sequencing data, to test InPACT. Specifically, we have incorporated RNA-seq data from the MicroArray/Sequencing Quality Control (MAQC) Universal Human Reference RNA (UHR) and human brain samples [9] (**Supplementary Table S1**). The polyA sites of these samples were experimentally identified using PolyA-seq, and we consider them as the ground truth for evaluation purposes (**Supplementary Figure S9**).

We have employed InPACT to identify IPA sites in these RNA-seq datasets and have utilized the matched experimental 3' end sequencing data as the ground truth to evaluate the performance of InPACT (**Figure 3, Supplementary Figure S9**). Additionally, as per suggestion from Reviewer #3, we have included APAIQ, a recently published tool, in our benchmark analysis. The results demonstrated that InPACT outperform other tools, including IPAFinder and APAIQ (**Figure 3, Supplementary Figure S9**). Notably, our revised manuscript incorporates diverse RNA-seq data derived from both cell lines and tissue samples to comprehensively evaluate the efficacy of InPACT.

5. InPACT should be compared to ground truth, which is polyA-seq, 3'-seq etc in this case and not just IPAFinder.

RESPONSE: We appreciate the comment provided by the reviewers. It is important to clarify that the A-seq and 3P-seq methods mentioned in our previous manuscript are both high-throughput sequencing techniques employed for the detection of polyA sites, similar to the polyA-seq and 3'-seq methods [7, 10, 11]. In our earlier version, we utilized A-seq and 3P-seq from HEK293 cells as a ground truth to assess the performance of InPACT and IPAFinder on RNA-seq data derived from the same cell line. Additionally, in the earlier version, we have also incorporated a long-read sequencing data of human small airway epithelial cells as a ground truth to evaluate the performance of InPACT. This approach is commonly employed in prior studies that focus on the development of computational methods for the identification of tandem polyA sites from RNA-seq data, such as IPAFinder, and APAIQ [12-14]. Notably, a recently published study that benchmarks APA detection methods also employed a similar strategy [15]. By adopting this strategy, we align our study with established practices in the

field, enabling a comprehensive evaluation of our method.

Moreover, we expanded our evaluation by including additional RNA-seq data from different tissue samples to assess the effectiveness of InPACT (**Supplementary Table S1, Supplementary Figure S9**). In this context, the polyA-seq data from the corresponding tissues were used as benchmarks for comparing InPACT with other publicly available methods for identifying IPA sites (**Supplementary Figure S9**). In summary, our revised manuscript incorporates 3' end sequencing experimental data (A-seq, 3P-seq, and PolyA-seq) [7, 10, 11] as well as long-read sequencing data to comprehensively evaluate the performance of InPACT.

6. Although the authors claim that InPACT can be used for characterization and quantification of IPA sites. There was no information on how quantification was done.

RESPONSE: We appreciate the comments provided by the reviewer. In the previous version of the manuscript, we briefly outlined the quantification process of InPACT following the “*Read module*” part of the Methods section. In the revised manuscript, we have expanded upon this description within a separate paragraph to elucidate how IPA quantification was achieved using InPACT in the context of RNA-seq data (**Page 21 Line 593-605**).

InPACT employs the *Sequence module* and *Read module* to accurately identify novel IPA sites and intronic terminal exons. Additionally, InPACT has the capability to assemble novel IPA isoforms based on reference annotation and search for the first in-frame stop codon within the isoform. The resulting detailed annotation of each IPA isoform is outputted in General Transfer Format (GTF) by InPACT. To measure the expression of all isoforms annotated in the GTF file generated by InPACT, we have incorporated Salmon, a fast and GC bias-aware quantification tool specifically designed for quantifying transcript-level expression from RNA-seq data [16]. Consequently, Salmon enables the estimation of the expression level of each IPA isoform from RNA-seq data.

In order to quantify the relative usage of IPA isoforms, InPACT calculates the relative expression an IPA isoform by comparing it to the total expression level of all isoforms within a gene. This metric, referred as IPA usage, allows for the assessment of the relative usage of IPA isoforms:

$$IPA\ usage = \frac{X_{ig}}{\sum_j X_{ig}}$$

Where g is a give gene, X_{ig} is the expression level of isoform i in gene g , measured in transcripts per million (TPM).

7. Throughout the manuscript, there are lot of details and information missing making it very difficult to follow through the different steps. For example, authors have reported p value <0.05 in line 179 without any mention of what statistical test they used etc. many more examples of missing information has been mentioned above.

RESPONSE: We express our sincere gratitude to the reviewer. In light of the valuable comments, we have endeavored to rectify all instances of missing information and offer explicit elucidations in our revised manuscript. With regards to the reported $P < 0.05$, the P -value was utilized to ascertain the statistical significance of the correlation between the expression levels of novel assembled IPA isoforms in two HEK293 RNA-seq replicates (**Page 8 Line 217**). To calculate the P -value, we employed a correlation test based on Pearson's correlation coefficient using the R function `cor.test()`. More specifically, the test statistics for Pearson's correlation coefficient can be expressed using the following formula.

$$t = \frac{r\sqrt{n-2}}{\sqrt{1-r^2}}$$

Where r is the Pearson's correlation coefficient, n is the number of observations. The P -value is $2 \times P(T > t)$ where T follows a t distribution with $n - 2$ degrees of freedom.

Reviewer #2 (Remarks to the Author):

In this manuscript, Liu and colleagues describe a new computational approach for identifying and quantifying the use of intronic polyA (IPA) sites. This method, termed InPACT, uses a variety of features, both learned from a reference genome sequence and observed in RNAseq data, to identify unannotated IPA sites. After demonstrating the power of the approach, they then use it to identify IPA sites that are differentially regulated upon monocyte activation with LPS. Finally, they apply their approach to a single-cell RNAseq dataset from fetal bone marrow, demonstrating that many of these new IPA sites are differentially used across cell types.

Overall, I found that the manuscript made a clear case for why this technique is superior to what is currently available. I believe that the method will be a valuable addition to the field. I have a few suggestions that may improve the manuscript.

RESPONSE: We extend our heartfelt appreciation to the reviewer for their diligent assessment and enlightening suggestions. In response, we have diligently incorporated the suggested revisions and conducted supplementary analyses to improve the manuscript. Important revisions have been highlighted in red in the revised version.

MAJOR COMMENTS

1. In figure 2E, the authors compare conservation scores of sequences surrounding previously annotated and newly identified IPA sites. This is fine, but it's hard to really interpret this result without a negative control sequence set. Yes, newly identified IPA sites are less conserved than previously annotated ones, which might be expected. A more interesting and interpretable question, though, would be to ask whether the newly identified sites are more conserved than expected (i.e. to other intronic sequences that may act as suitable negative controls). The new IPA sites may be expected to be more conserved than these negative control sites, and if so, this would be additional evidence of their existence and function.

RESPONSE: We express our gratitude to the reviewer for their valuable suggestions, which we find to be highly constructive. We concur that the inclusion of a negative control in our comparative analysis of sequence conservation would significantly bolster the confidence in our findings. In response to this comment, we have taken the following steps: First, we randomly selected a set of genomic sites from annotated intronic regions, ensuring that the size of this negative control group matched that of analyzed IPA sites (**Page 22 Line 610-612**). Subsequently, we calculated the evolutionary conservation of these negative control sites like the analyzed IPA sites. Our observations revealed that the newly identified IPA sites exhibit a higher degree of conservation compared to the negative control group, although a lesser extent than the annotated polyA sites (**Figure 2G**). These results provide further support for the authenticity of the IPA sites identified by InPACT.

2. In figure 4D, the authors attempt to show that changes in IPA usage do not have a noticeable

effect on gene (RNA) expression as there is little overlap between the genes that display differential IPA usage and those that are differentially expressed. This analysis could be improved. For example, if you just look at genes that are differentially expressed, you are combining those that increase with expression and those that decrease in expression. If the connection between IPA usage and gene expression is directionally correlated, then you might expect a result where, for example, usage of an upstream IPA site tends to increase gene expression while usage of a downstream site decreases gene expression. This exact result has actually been observed before (Goering et al, BMC Genomics 2021, figure S4B and S4D). Both thresholding on a significance cutoff for differential expression and IPA usage and lumping all gene expression changes together may obscure this result.

RESPONSE: We extend our sincere appreciation to the reviewer for the invaluable suggestion on improving the analysis concerning the association between IPA usage and gene expression. We agree with the reviewer that a more nuance examination is imperative to capture directional correlations.

In order to investigate the relationship between IPA usage and gene expression, we conducted a meticulous comparative analysis of the changes in gene expression and IPA usage between the treatment and control groups (**Figure 4D**). Our observations revealed a rather weak correlation between changes in IPA usage and gene expression, as indicated by Spearman's correlation ($\rho = -0.079$, $P = 0.0049$) (**Figure 4D, Page 13 Line 349-352**). These findings suggested that IPA may represent a distinct regulatory layer, functioning relatively independently of gene expression

MINOR COMMENTS

1. In a few places, the authors fail to cite work that has demonstrated clear function for IPA. This can perhaps partially be attributed to the fact that earlier literature on the subject sometimes referred to these isoforms as ALE (alternative last exons) rather than IPA. However, the two terms refer to the same thing. Although IPA isoforms differ in their C-terminal coding regions, they also differ in 3' UTR content (which is not discussed in the manuscript). One consequence of this difference in 3' UTR content is that transcripts that use different IPA (ALE) sites are often differentially localized within cells (Taliaferro et al, Mol Cell 2016). The reason I

bring this up is that the introduction reads like the function of IPA is unknown.

RESPONSE: We express our gratitude to the reviewer for providing valuable insights. In response to the comment, we have made revisions to the manuscript to include the mention that IPA isoforms are also known as alternative last exons (ALE) (**Page 3 Line 59**). In the earlier version, we have expanded up the functions of IPA in various biological processes and pathological conditions, particularly in generating truncated proteins that lack C-terminal domains. Notably, aberrant IPA events have been shown to produce truncated proteins that inactivate tumor suppressor genes such as *DICER*, *FOXN3*, and *MGA* [17]. We concur with the reviewer's comments that IPA can also give rise to isoforms with distinct 3' UTR contents, thereby influencing the subcellular localization of RNA molecules. To support this notion, we have incorporated the findings from Taliaferro et al., who investigated subcellular transcriptomes in primary mouse cortical neurons [18] (**Page 4 Line 77-80**). They examined RNA localization at the isoform level and identified numerous alternative 3' UTRs associated with mRNA localization to neurites. Notably, gene-distal alternative last exon isoforms were found to preferentially localize to neurites [18]. We have diligently revised the manuscript to incorporate these pivotal findings, providing appropriate references. We express our sincere gratitude to the reviewer for the insightful comments.

2. The authors state that currently available APA analysis tools are not well-suited for IPA quantification. This is true for the tools listed, but at least one that was not listed (LABRAT, Goering et al, BMC Genomics 2021), can quantify differential IPA (ALE) isoform usage. Related to Major Comment 2, IPA usage again was shown to be connected to transcript abundance. A major advantage of InPACT over LABRAT is InPACT's ability to identify new IPA isoforms. This is something LABRAT cannot do.

A much older software package aimed at quantifying alternative splicing, MISO, also has the ability to quantify ALE usage, but again cannot identify new isoforms.

RESPONSE: We express our gratitude to the reviewer for providing insightful comments. In response, we have made revisions to our manuscript to include additional tools for quantifying IPA, namely LABRAT and MISO [19, 20] (**Page 4 Line 88-90**). LABRAT has demonstrated its utility in determining the relative usage of APA sites and elucidating the co-regulation between

tandem UTRs and alternative last exons [19]. Nevertheless, it is important to note that LABRAT relies on existing genome annotation and lacks the capability to detect novel IPA isoforms. Similarly, MISO primarily serves as a statistical model for estimating the expression levels of alternative spliced exons and does not possess the capacity to identify novel isoforms [20]. We have diligently revised the manuscript to include these two tools, providing appropriate references.

3. This is just a suggestion, and not something that should be required for publication, but given InPACT's ability to identify new IPA isoforms, it could be interesting to look at what InPACT says in datasets in which U1 snRNP function has been inhibited. There is quite a bit of literature about U1's ability to block the usage of cryptic, upstream, often intronic polyA sites (so-called "telescripting", for review see PMID 30709878). If for example, InPACT detected many new IPA isoforms in samples where U1 was inhibited, this would be even further evidence of the high performance of InPACT and would really highlight its ability to find new IPA sites.

RESPONSE: We express our gratitude to the reviewer for their insightful comments. The interaction between U1 small nuclear ribonucleoprotein (snRNP) and factors involved in cleavage and polyadenylation plays a crucial role in regulating premature 3' end cleavage and polyadenylation (PCPA) by binding to cryptic intronic polyA sites [21-23]. This process, known as telescripting, is essential for ensuring complete transcription and serves as a general mechanism for controlling transcription elongation [21-23]. We agreed with the reviewer's suggestion to investigate the impact of U1 snRNP inhibition on IPA using InPACT in relevant datasets (**Page 9 Line 229-242**).

Consequently, we applied the InPACT to an RNA-seq dataset obtained from HeLa cell treated with antisense morpholino oligonucleotide (AMO) targeting U1 and a control group [23] (**Page 9 Line 229-242, Supplementary Figure S7**). Previous studies have confirmed that U1 AMO disrupts the structure of U1 snRNP, leading to the promotion of intronic PCPA [23]. The dataset used in this study was obtained from GEO under the accession number GSE193200 [23]. Our analysis identified a total of 767 novel IPA events, including 254 skipped and 513 composite IPA events, in HeLa cells treated with U1 AMO. In contrast, the control group

exhibited only 151 IPA events, consisting of 123 skipped and 28 composite IPA events. Furthermore, we investigated the dynamic usage of IPA sites in HeLa cells treated with U1 AMO compared to the control group. Our observations revealed a significant increase in the usage of IPA sites following U1 AMO treatment in HeLa cells (Wilcoxon rank sum test, $P < 2.2e-16$). These findings obtained through applying InPACT to RNA-seq data of U1 inhibition are consistent with the telescripting activity of U1 snRNP. In summary, our results provided additional compelling evidence supporting the efficacy of InPACT.

Reviewer #3 (Remarks to the Author):

Liu et al present a computational method for accurate characterization of intronic polyadenylation from standard RNA-seq data. While they showed that InPACT is a powerful tool, they ignore the most relevant tool called APAIQ that recently published in Genome Research for the direct comparison. This emphasizes a fact that InPACT is not the first method utilizing the synergistic effect of RNA-seq read coverage and DNA sequence on IPA site identification and should be mentioned. In addition, a comprehensive comparison between InPACT and APAIQ should be carried out before the consideration of publication in Nature Communications. Therefore, I would suggest 'major revision' for this manuscript.

RESPONSE: We extend our sincere appreciation to the esteemed reviewer for the valuable insights and suggestion pertaining to our manuscript. These comments have greatly improved our manuscript. We have diligently incorporated all the suggested revisions and made substantial revisions in the manuscript, including the comparison between InPACT and APAIQ. The important changes are highlighted in red for you to access easily.

Major comments:

1. As we know, there exists synergistic effect of RNA-seq read coverage and DNA sequence on PAS identification. IPAFinder performed de novo IPA identification and quantification from standard RNA-seq data based on "change point" model, which also have been used to analyse tandem APA events by DaPars or PAQR. Given that IPAFinder only considered RNA-seq read coverage information, thus IPAFinder is sensitive to coverage bias of mRNA 3' end, which may

influence the identification of accurate position of IPA sites. InPACT developed in this manuscript considered more features than IPAFinder, which included both RNA-seq read coverage information and genomic information associated with mRNA 3' end processing. Thus, InPACT deservedly better than IPAFinder. I think that authors should refer to this point in the main text.

RESPONSE: We express our gratitude to the reviewer for their insightful remarks concerning the distinction between InPACT and IPAFinder. IPAFinder employs the “change point” model to detect and quantify IPA events from standard RNA-seq in a *de novo* manner. However, it solely relies on RNA-seq read coverage information renders it vulnerable to potential biases in mRNA 3' end coverage, which could consequently affect the precise identification of IPA sites. In contrast, InPACT incorporates additional features beyond read alignment information, including genomic sequence information associated with mRNA 3' end cleavage and polyadenylation. Consequently, it is expected that InPACT will outperform IPAFinder in terms of accurately identifying IPA sites. We have made the revisions to the manuscript to highlight this aspect in the revised version (**Page 9 Line 254-259**).

2. Authors didn't mention statistics model about detecting significantly differentially used IPA sites in the method section. I think that the appropriate statistics model is very important for analyzing small number of samples (5 controls vs 5 cases).

RESPONSE: We express our gratitude to the reviewer for their constructive comments. We agree with the reviewer that it is important to select an appropriate statistical model for analyzing datasets with a limited number of replicates, a common occurrence in high-through sequencing experiments. In our previous manuscript, we have acknowledged the utilization of DRIMSeq as a tool for identifying significantly differentially used IPA sites in the Methods section.

In our analysis of monocytes' RNA-seq data, the expression level of all isoforms can be estimated by InPACT. Then, we employed DRIMSeq to perform differential transcript usage analysis between the control and LPS stimulated groups [24] (**Page 12 Line 336-339, Page 24 Line 679-681**). The DRIMSeq software has been extensively utilized in previous studies for statistical analysis of differential transcript usage [25-28]. DRIMSeq employs a statistical

framework based on the Dirichlet-multinomial distribution, which allows for the identification of changes in isoform usage between conditions. The Dirichlet-multinomial model accounts for the differential expression without losing information about overall gene abundance. This model treats isoform expression as a multivariate response and can handle scenarios with limited samples by allowing flexible estimation of overdispersion. DRIMSeq tackle the issue of obtaining reliable model parameters, when only a small number of replicates are available, by implementing an empirical Bayes approach to leverage shared information. This approach is akin to other successful methods employed in negative binomial frameworks [29, 30].

We extend our appreciation to the reviewer for suggesting that we mention the statistical model employed in the analysis of differential isoform usage. We have revised the manuscript to provide a detailed description in the differential IPA analysis of monocytes' RNA-seq data (**Page 12 Line 336-339, Page 24 Line 679-681**).

3. Authors didn't mention APAIQ method (published by a recent Genome Research paper (PMID: 37117035)), which could be used to identify both UTR APA and IPA sites. Most importantly, APAIQ also integrates RNA-seq read coverage information with genomic sequence through CNN. Thus, InPACT developed here was not the first method utilizing the synergistic effect of RNA-seq read coverage and DNA sequence on PAS identification. I think authors should compare their InPACT with APAIQ, which is more appropriate than comparison with IPAFinder. The direct comparison between InPACT and APAIQ should be put into the central position of the result section. Comparison to IPAFinder could be moved to the Discussion section or Supplementary Materials.

RESPONSE: We express our gratitude to the reviewer for the constructive comments. We concur with the reviewer's suggestion to compare InPACT with APAIQ, a recently published method that leverages the combined effect of DNA sequence and RNA-seq read coverage for the identification of polyA sites. In response, we have included a comparative analysis between InPACT and APAIQ in the revised version of our manuscript (**Figure 3**). It should be noted that while APAIQ can detect IPA sites from RNA-seq data by integrating RNA-seq read coverage information with genomic sequence, it is not specifically designed for IPA analysis and therefore cannot differentiate between skipped and composite IPA events [14]. However,

detailed annotation of IPA events is important for further investigation of the biological functions of IPA.

Initially, we examined the enrichment of the canonical polyA signal (AAUAAA) for IPA sites identified by InPACT, APAIQ, and IPAFinder. The results demonstrated a significant enrichment of the canonical polyA signal for IPA sites by InPACT and APAIQ (**Figure 3A**). Furthermore, we also investigated the nucleotide compositions surrounding IPA sites identified by each tool. The nucleotide profiles around IPA identified by InPACT and APAIQ closely resembled those of annotated polyA sites (**Supplementary Figure S8**).

In addition, we utilized experimental 3'-end sequencing data to assess the performance of each tool. This evaluation incorporated sequencing data from three different 3'-end sequencing protocols: 3P-seq, A-seq, and PolyA-seq [7, 10, 11]. The observations showed that InPACT outperforms APAIQ in accurately identifying IPA sites from conventional RNA-seq data (**Page 10 Line 270-286, Figure 3B, C, Supplementary Figure S9**). Furthermore, the observations were also revealed by the RNA-seq data from human small airway epithelial cells, which were evaluated against matched long-read Iso-seq data (**Figure S3D**). These results consistently demonstrated that InPACT outperforms APAIQ in accurately identifying IPA sites from conventional RNA-seq data.

Moreover, we employed simulated RNA-seq data to further evaluate the performance of each tool. These simulations encompassed a range of sequencing coverage, ranging from 10X to 50X. Our meticulous analysis revealed that as the sequencing depth increased, InPACT consistently outperforms APAIQ in identifying IPA events (**Figure 3G, H**). Notably, at a sequencing coverage of 50X, InPACT successfully recovered approximately 90% of IPA sites. Additionally, the comparative analysis extended to the quantification of IPA events using simulated RNA-seq data. InPACT exhibited a lower error rate compared to APAIQ in quantifying IPA events (**Figure 3I**). To further investigate the impact of sequencing depth on quantification accuracy, we conducted analyses using simulated RNA-seq data with coverage ranging from 10X to 50X (**Figure 3I, Supplementary Figure S12**). The results consistently demonstrated that InPACT outperforms APAIQ in accurately quantifying IPA events from RNA-seq data.

In response to the constructive comments provided by the reviewer, we have compared

InPACT with APAIQ, and the results collectively demonstrate that InPACT outperforms APAIQ in identifying and quantifying IPA from conventional RNA-seq data. The reviewer suggested moving the comparison with IPAFinder to the Discussion section or Supplementary Materials. However, as IPAFinder is a tool specifically designed for IPA analysis published recently, we believe it necessary to present the comparison in the main figures. We extend our sincere appreciation to the reviewers for their insightful suggestions, which have significantly enhanced the overall quality and clarity of our manuscript.

4. The authors didn't mention deep learning models APARENT (PMID: 31178116) and DeepPASS (PMID: 34376223). APARENT model could predict polyA site strength based on genomic sequence. DeepPASS model could precisely identify true polyA site based on genomic sequence with high sensitivity and accuracy, which further be used to analyze differential transcript expression at single-cell resolution. Given that the result about that applying sequence module of InPACT on PolyASite 2.0 prediction obtain an AUROC of 0.794, I have concerns about the performance of CNN model of InPACT. I think performing the comparison between InPACT with APARENT and DeepPASS is necessary.

RESPONSE: We thank the reviewer for the constructive comments. We agree with the reviewer's suggestion to compare the *Sequence module* of InPACT with other existing deep learning models. In line with this suggestion, we have conducted a comprehensive comparative analysis involving three recently published deep learning models: APARENT, DeepPASS, and DeepPASTA [1, 5, 6] (**Page 6 Line 144-150**).

The Sequence module of InPACT was trained a curated set of polyA sites annotated in Refseq database. In contrast, DeepPASS was trained on a stringent set of polyA sites annotated in all three databases of PolyA_DB 3, PolyASite 2.0 and the PolyA-seq paper [1, 7], or in the GENCODE annotation. The APARENT model was trained using a massively parallel reporter assay that involved over 3 million synthetic APA reporter constructs [5]. Lastly, the DeepPASTA model was trained on the polyA sites annotated the PolyA-seq paper [6, 7].

We incorporated three different polyA site annotation for the purpose of evaluation, namely GENCODE, PolyA_DB 3, PolyASite 2.0. In response to the suggestion put forth by Reviewer #1, we excluded the polyA sites that overlapped with the polyA sites used for training

our model on Refseq. This step was taken to ensure an appropriate assessment of our model. The results of our evaluation revealed that our model achieved effective predictions for the testing polyA sites, with AUROC values of 0.954 for GENCODE, 0.920 for PolyA_DB 3, and 0.794 for PolyASite 2.0 (**Supplementary Figure S1B-D**). Comparing these results with the performance of other models, we observed that our model's overall performance is either better or comparable. Specifically, when compared to DeepPASS (GENCODE: AUROC = 0.989, PolyA_DB 3: AUROC = 0.926, PolyASite 2.0: AUROC = 0.840), APARENT (GENCODE: AUROC = 0.944, PolyA_DB 3: AUROC = 0.852, PolyASite 2.0: AUROC = 0.800), and DeepPASTA (GENCODE: AUROC = 0.913, PolyA_DB 3: AUROC = 0.844, PolyASite 2.0: AUROC = 0.791), our model's performance remains competitive (**Supplementary Figure S1B-D**). Notably, we observed that all models had relatively lower performance on PolyASite 2.0 compared to other testing sets of polyA sites (**Supplementary Figure S1D**). DeepPASS, however, exhibited slightly better performance on PolyASite 2.0. It is important to acknowledge that the superior performance of DeepPASS on PolyASite 2.0 can be attributed to its training on a comprehensive set of polyA sites from multiple databases, including the PolyASite 2.0. In summary, our findings strongly support the notion that the Sequence module of InPACT demonstrates a high level of confidence in identifying poly sites based on genomic sequences.

5. Authors have performed some analyses about applying InPACT on scRNA-seq data. Nowadays, a large amount of scRNA-seq data was accumulated in the public databases. I wonder whether InPACT is expandable on the other type (3' tag but not full-length) of widely used scRNA-seq data (such as 10x Genomics), which will extend the application range of InPACT and attract more interest of potential users.

RESPONSE: We acknowledge and appreciate the reviewer's suggestion regarding the potential application of InPACT on 3' tag-based single-cell RNA sequencing (scRNA-seq) data. We would like to mention that the previous manuscript has included this issue in the Discussion section (**Page 16 Line 450-457**). The characterization of IPA from 3' tag-based scRNA-seq data is an intriguing direction for future investigations. However, it is essential to emphasize that InPACT is primarily designed for analyzing traditional bulk RNA sequencing data and full-length scRNA-seq data, such as Smart-seq2, and may not be suitable for 3' tag-

based scRNA-seq platforms like 10x Genomics. The core component of InPACT, the *Read module*, is a sample-specific machine-learning model trained to distinguish terminal exons from internal exons and background regions. To ensure the model's accuracy, it relies on read alignments distributed across three distinct types of regions. However, 3' tag-based scRNA-seq data may not provide complete coverage of terminal exons and internal exons, posing challenges for the accurate identification of terminal exons by the model.

While we appreciate the suggestion to explore the application of InPACT on 3' tag-based scRNA-seq data, we would like to clarify that this is currently beyond the scope of InPACT's capabilities. Nevertheless, we recognize the significance of addressing this potential avenue for future research. Notably, several computational methods, including Sierra, SAPAS and SCAPTURE [1, 31], have been developed to analyze APA using 3' tag-based scRNA-seq data, which can be employed for IPA analysis. It is worth mentioning that these methods are limited to identifying IPA sites and cannot assemble the IPA isoforms. We express our gratitude to the reviewer for their valuable input, which aids in elucidating the current limitations of InPACT and identifying areas for potential expansion.

6. The authors should perform experimental validations for at least ten genes in diverse cell types or biological processes such as KEK293 cells (Figure 2), human small airway epithelial cells (Figure 3), monocytes activation (Figure 4), and human fetal bone marrow cells (Figure 5). For a methodology paper, ten genes are reasonable number for the experimental validation. In Fig 2A, 2B, 2I and 2J, RT-PCR and Sanger sequencing should be performed to demonstrate the reliability of IPA sites in ZNF771, TERF2, PIGL and CHRNA5 genes. In Fig. 3E and 3F, RT-PCR and Sanger sequencing should be performed to demonstrate the reliability of IPA sites in HPS1 and CDC23 genes. In Fig. 4F, RT-PCR and Sanger sequencing should be performed to demonstrate the reliability of IPA sites in ARHGAP24 and more other genes. In Fig. 5C and 5D, RT-PCR and Sanger sequencing should be performed to demonstrate the cell-type specificity of IPA sites in SRP68 and HMGCL genes. Results regarding the B cell-specific IPA site in SRP68 and Basophil, DC2, Eosinophil-specific IPA sites in HMGCL should be shown.

RESPONSE: We appreciate the reviewer's suggestion regarding the need for experimental

validation of the identified IPA sites. However, it is important to clarify that our manuscript primarily focuses on the development and application of InPACT for characterizing IPA using RNA-seq data. Nonetheless, we recognize the significance of experimental validation in reinforcing the reliability of InPACT. In response to the reviewer's suggestion, we have performed experimental validation on a set of candidate IPA events. The experimental validation was carried out using the 3'-Rapid Amplification of cDNA Ends (3'-RACE) technique, which involves selective amplification of the 3' ends of cDNAs followed by Sanger sequencing of individual clones (**Page 25 Line 702-710**). This classical method allows for the determination of polyA sites of RNAs and the acquisition of 3' end sequences adjacent to known sequences.

In the case of HEK293 cells (**Figure 2**), we have chosen 15 candidate IPA sites for experimental validation to demonstrate the reliability of InPACT (**Page 7 Line 191-197, Figure 2, Supplementary Figure S6**). These sites include IPA sites within the *ZNF771*, *TERF2*, *PIGL* and *CHRNA5* genes, as depicted in **Figure 2**. Summarizing the experimental results, we found that 10 out of the 15 candidate IPA sites were experimentally confirmed within a distance of 10 nucleotides from the predicted positions, while the remaining 5 sites were confirmed within a distance of about 40 nucleotides. Therefore, all candidate IPA sites were successfully confirmed using 3'-RACE in HEK293 cells (**Figure 2, Supplementary Figure S6**).

For monocytes (**Figure 4**), we have selected 4 candidate IPA sites identified by InPACT for experimental validation (**Page 13 Line 363-368, Figure 4, Supplementary Figure S14**). These sites include the IPA site in the *ARHGAP24* gene, as shown in **Figure 4**. Upon summarizing the experimental results, we successfully validated three candidate IPA sites in LPS stimulated cells, while one candidate IPA site failed validation, possibly due to its relatively low expression level (**Figure 4, Supplementary Figure S14**). Nonetheless, these experimental results provide evidence for the presence of the predicted IPA sites, thus affirming the reliability of InPACT.

In the case of human small airway epithelial cells (**Figure 3**), we have utilized matched PacBio Iso-seq data from human small airway epithelial cells to demonstrate the presence of the candidate IPA sites in the previous manuscript. Iso-seq is renowned for its ability to capture high-quality, full-length transcript sequences in a single sequencing read, thereby providing

reliable isoform information without the requirement for transcript reconstruction [32]. Furthermore, the Iso-seq method has been acknowledged as the ground truth for evaluating APA detection methods in a recently published paper [15]. By leveraging the matched Iso-seq data from human small airway epithelial cells, we have successfully confirmed the presence of 3' ends for candidate IPA sites in *HPS1* and *CDC23* (**Supplementary Figure S10, Page 11 Line 298-300**). The compelling evidence furnished by the Iso-seq data substantiates the reliability of these candidate IPA sites, thereby obviating the need for additional experimental validation.

According to the reviewer's suggestion, we have tried our best to conduct experimental validation to demonstrate the reliability of IPA sites identified by InPACT. Regarding human fetal bone marrow (Figure 5), we concur with the reviewer's assertion regarding the importance of experimental validation. However, due to current limitations, obtaining clinical samples of human fetal bone marrow within the recent timeframe has proven to be unfeasible. Although validation in human fetal bone marrow was not carried out, we firmly believe that the additional experimental results presented herein provide compelling evidence supporting the authenticity and accuracy of the IPA sites identified by InPACT. We express our sincere appreciation for the reviewer's constructive suggestion, which further corroborates the reliability of InPACT.

References

1. Li GW, Nan F, Yuan GH, Liu CX, Liu X, Chen LL, Tian B, Yang L: **SCAPTURE: a deep learning-embedded pipeline that captures polyadenylation information from 3' tag-based RNA-seq of single cells.** *Genome Biol* 2021, **22**:221.
2. Frankish A, Diekhans M, Jungreis I, Lagarde J, Loveland JE, Mudge JM, Sisu C, Wright JC, Armstrong J, Barnes I, et al: **Gencode 2021.** *Nucleic Acids Res* 2021, **49**:D916-D923.
3. Wang R, Nambiar R, Zheng D, Tian B: **PolyA_DB 3 catalogs cleavage and polyadenylation sites identified by deep sequencing in multiple genomes.** *Nucleic Acids Res* 2018, **46**:D315-D319.
4. Herrmann CJ, Schmidt R, Kanitz A, Artimo P, Gruber AJ, Zavolan M: **PolyASite 2.0: a consolidated atlas of polyadenylation sites from 3' end sequencing.** *Nucleic Acids Res* 2020, **48**:D174-D179.
5. Bogard N, Linder J, Rosenberg AB, Seelig G: **A Deep Neural Network for Predicting and Engineering Alternative Polyadenylation.** *Cell* 2019, **178**:91-106 e123.
6. Arefeen A, Xiao X, Jiang T: **DeepPASTA: deep neural network based polyadenylation site analysis.** *Bioinformatics* 2019, **35**:4577-4585.
7. Derti A, Garrett-Engle P, Macisaac KD, Stevens RC, Sriram S, Chen R, Rohl CA, Johnson JM, Babak T: **A quantitative atlas of polyadenylation in five mammals.** *Genome Res* 2012, **22**:1173-1183.
8. Liu N, Dai Q, Zheng G, He C, Parisien M, Pan T: **N(6)-methyladenosine-dependent RNA structural switches regulate RNA-protein interactions.** *Nature* 2015, **518**:560-564.
9. Rapaport F, Khanin R, Liang Y, Pirun M, Krek A, Zumbo P, Mason CE, Socci ND, Betel D: **Comprehensive evaluation of differential gene expression analysis methods for RNA-seq data.** *Genome Biol* 2013, **14**:R95.
10. Martin G, Gruber AR, Keller W, Zavolan M: **Genome-wide analysis of pre-mRNA 3' end processing reveals a decisive role of human cleavage factor I in the regulation of 3' UTR length.** *Cell Rep* 2012, **1**:753-763.
11. Jan CH, Friedman RC, Ruby JG, Bartel DP: **Formation, regulation and evolution of *Caenorhabditis elegans* 3'UTRs.** *Nature* 2011, **469**:97-101.
12. Xia Z, Donehower LA, Cooper TA, Neilson JR, Wheeler DA, Wagner EJ, Li W: **Dynamic analyses of alternative polyadenylation from RNA-seq reveal a 3'-UTR landscape across seven tumour types.** *Nat Commun* 2014, **5**:5274.
13. Zhao Z, Xu Q, Wei R, Wang W, Ding D, Yang Y, Yao J, Zhang L, Hu Y-Q, Wei G, Ni T: **Cancer-associated dynamics and potential regulators of intronic polyadenylation revealed by IPAFinder using standard RNA-seq data.** *Genome Research* 2021, **31**:2095-2106.
14. Long Y, Zhang B, Tian S, Chan JJ, Zhou J, Li Z, Li Y, An Z, Liao X, Wang Y, et al: **Accurate transcriptome-wide identification and quantification of alternative polyadenylation from RNA-seq data with APAIQ.** *Genome Res* 2023, **33**:644-657.
15. Shah A, Mittleman BE, Gilad Y, Li YI: **Benchmarking sequencing methods and tools that facilitate the study of alternative polyadenylation.** *Genome Biol* 2021, **22**:291.
16. Patro R, Duggal G, Love MI, Irizarry RA, Kingsford C: **Salmon provides fast and bias-aware quantification of transcript expression.** *Nat Methods* 2017, **14**:417-419.
17. Lee S-H, Singh I, Tisdale S, Abdel-Wahab O, Leslie CS, Mayr C: **Widespread intronic polyadenylation inactivates tumour suppressor genes in leukaemia.** *Nature* 2018,

- 561:127-131.
18. Taliaferro JM, Vidaki M, Oliveira R, Olson S, Zhan L, Saxena T, Wang ET, Graveley BR, Gertler FB, Swanson MS, Burge CB: **Distal Alternative Last Exons Localize mRNAs to Neural Projections.** *Mol Cell* 2016, **61**:821-833.
 19. Goering R, Engel KL, Gillen AE, Fong N, Bentley DL, Taliaferro JM: **LABRAT reveals association of alternative polyadenylation with transcript localization, RNA binding protein expression, transcription speed, and cancer survival.** *BMC Genomics* 2021, **22**:476.
 20. Katz Y, Wang ET, Airoidi EM, Burge CB: **Analysis and design of RNA sequencing experiments for identifying isoform regulation.** *Nat Methods* 2010, **7**:1009-1015.
 21. Venters CC, Oh JM, Di C, So BR, Dreyfuss G: **U1 snRNP Telescripting: Suppression of Premature Transcription Termination in Introns as a New Layer of Gene Regulation.** *Cold Spring Harb Perspect Biol* 2019, **11**.
 22. So BR, Di C, Cai Z, Venters CC, Guo J, Oh JM, Arai C, Dreyfuss G: **A Complex of U1 snRNP with Cleavage and Polyadenylation Factors Controls Telescripting, Regulating mRNA Transcription in Human Cells.** *Mol Cell* 2019, **76**:590-599 e594.
 23. Feng Q, Lin Z, Deng Y, Ran Y, Yu R, Xiang AP, Ye C, Yao C: **The U1 antisense morpholino oligonucleotide (AMO) disrupts U1 snRNP structure to promote intronic PCPA modification of pre-mRNAs.** *J Biol Chem* 2023, **299**:104854.
 24. Nowicka M, Robinson MD: **DRIMSeq: a Dirichlet-multinomial framework for multivariate count outcomes in genomics.** *F1000Res* 2016, **5**:1356.
 25. Glinos DA, Garborcauskas G, Hoffman P, Ehsan N, Jiang L, Gokden A, Dai X, Aguet F, Brown KL, Garimella K, et al: **Transcriptome variation in human tissues revealed by long-read sequencing.** *Nature* 2022, **608**:353-359.
 26. Hallegger M, Chakrabarti AM, Lee FCY, Lee BL, Amalietti AG, Odeh HM, Copley KE, Rubien JD, Portz B, Kuret K, et al: **TDP-43 condensation properties specify its RNA-binding and regulatory repertoire.** *Cell* 2021, **184**:4680-4696 e4622.
 27. Sugimoto Y, Ratcliffe PJ: **Isoform-resolved mRNA profiling of ribosome load defines interplay of HIF and mTOR dysregulation in kidney cancer.** *Nat Struct Mol Biol* 2022, **29**:871-880.
 28. Yu T, Cazares O, Tang AD, Kim HY, Wald T, Verma A, Liu Q, Barcellos-Hoff MH, Floor SN, Jung HS, et al: **SRSF1 governs progenitor-specific alternative splicing to maintain adult epithelial tissue homeostasis and renewal.** *Dev Cell* 2022, **57**:624-637 e624.
 29. McCarthy DJ, Chen Y, Smyth GK: **Differential expression analysis of multifactor RNA-Seq experiments with respect to biological variation.** *Nucleic Acids Res* 2012, **40**:4288-4297.
 30. Robinson MD, McCarthy DJ, Smyth GK: **edgeR: a Bioconductor package for differential expression analysis of digital gene expression data.** *Bioinformatics* 2010, **26**:139-140.
 31. Patrick R, Humphreys DT, Janbandhu V, Oshlack A, Ho JWK, Harvey RP, Lo KK: **Sierra: discovery of differential transcript usage from polyA-captured single-cell RNA-seq data.** *Genome Biol* 2020, **21**:167.
 32. Rhoads A, Au KF: **PacBio Sequencing and Its Applications.** *Genomics Proteomics Bioinformatics* 2015, **13**:278-289.

Reviewer #2 (Remarks to the Author):

The authors have addressed my concerns.

Reviewer #3 (Remarks to the Author):

The authors have addressed most of my concerns and the revision is greatly improved. Two minor points:

1. page 21: it would be nice to change "relative expression IPA isoforms" to "relative expression an IPA isoform".

2. The IPA analysis of HEK293 cells between two replicates shows a relative low overlap, with about 50% consistency (sup figure S5). In principle, two replicates should have good consistency, regardless gene expression or RNA processing. I wonder how to explain the clear difference of these two replicates, due to the data or computational method? An explanation is required, possibly in discussion or somewhere else.

Reviewer #4 (Remarks to the Author):

In this revised manuscript, the authors Liu et al. proposed a deep learning based Intronic polyA detection model named InPACT which can identify novel Intronic polyA (IPA) events, as well as measure the quantification of the newly generated IPA isoforms from human RNA-seq data. They used two distinct modules: Sequence module and Read module to learn the sample-wise sequence patterns and read alignment information. Some major contributions of this work include:

- The accuracy of the modules is tested for several human samples and is validated with various datasets, like replicated samples, nucleotide composition of the surrounding region and ribosome profiling etc. They also measured the differential IPA usage for human HeLa cells: control vs U1-AMO treated, and monocyte activated samples: treated vs untreated. In all the measures, InPACT performs very well in detecting the novel IPA events and the differential ones.
- InPACT showed significant improvement in performance compared with 2 other baselines: IPAFinder and APAIQ. Baselines are compared against the truth values using 3'-end-seq, PacBio long read and simulated samples.
- The authors also showed how InPACT can be utilized for the detection of potential IPA events on single cell RNA-seq data.

In response to the feedback made by reviewer #1, the authors Liu et al. made substantial changes to their manuscript. They have added details about the training/testing data, including the generation of positive and negative samples, also described the sequence module and the read module in a more comprehensive way. They excluded the overlapping IPA sites from the different datasets which have been used in training and testing: RefSeq, GENCODE, PolyA_DB3 and PolyASite 2.0, to prevent the overfitting of results. Three other deep learning tools (DeepPASS, APARENT and DeepPASTA) have been mentioned and compared with the sequence module of InPACT, but no significant differences have been found in their way of selecting the annotated IPA sites from different databases. In accordance with the comment from reviewer #1, the authors also incorporated several human cell-lines and showed how InPACT outperformed the baselines (IPAFinder and APAIQ). Therefore, the quantification of the new IPA isoforms and their relative usage are also shown in the manuscript. Overall, the authors have adequately addressed most of Reviewer 1's concerns.

Point-by-point address of the comments raised by the reviewers for manuscript NCOMMS-23-25772A

We express our sincere gratitude to the esteemed editor and diligent reviewers for the comprehensive assessment of our revised manuscript. The valuable insights have greatly enhanced the quality of our paper. Please see below for a point-by-point response to the reviewers' comments.

Reviewer #2 (Remarks to the Author):

The authors have addressed my concerns.

RESPONSE: We express our sincere gratitude for your time and efforts in reviewing our manuscript, as well as for your insightful comments.

Reviewer #3 (Remarks to the Author):

The authors have addressed most of my concerns and the revision is greatly improved. Two minor points:

RESPONSE: We extend our gratitude for your time and efforts in reviewing our manuscript, as well as your constructive comments.

1. page 21: it would be nice to change "relative expression IPA isoforms" to "relative expression an IPA isoform".

RESPONSE: Thank you for the constructive comment. We have revised the manuscript accordingly.

2. The IPA analysis of HEK293 cells between two replicates shows a relative low overlap, with about 50% consistency (sup figure S5). In principle, two replicates should have good consistency, regardless gene expression or RNA processing. I wonder how to explain the clear difference of these two replicates, due to the data or computational method? An explanation is required, possibly in discussion or somewhere else.

RESPONSE: We thank the reviewer for pointing out this issue. To address this, we examined

the gene expression levels of overlapped and non-overlapped sites in the two replicates. Our findings indicate that non-overlapped sites have lower gene expression levels compared to overlapped sites (Supplementary Figure 5d). The lower gene expression abundance can result in insufficient read coverage of the non-overlapped IPA isoforms, leading to increased unexpected noise. Hence, the relatively low read coverage of these non-overlapped IPA isoforms may contribute to the moderate consistency observed between two replicates. Additionally, we assessed the consistency between two replicates using InPACT, IPAFinder and APAIQ. The results revealed an overlap rate of approximately 50% for InPACT, 30% for IPAFinder and 20% for APAIQ, with InPACT showing the highest overlap rate. While all methods consider read coverage information, achieving consistent distribution of reads across replicates for each isoform, particularly for those with low expression levels, can pose a challenge. We have added the explanation in the corresponding part of Results section.

Reviewer #4 (Remarks to the Author):

In this revised manuscript, the authors Liu et al. proposed a deep learning based Intronic polyA detection model named InPACT which can identify novel Intronic polyA (IPA) events, as well as measure the quantification of the newly generated IPA isoforms from human RNA-seq data. They used two distinct modules: Sequence module and Read module to learn the sample-wise sequence patterns and read alignment information. Some major contributions of this work include:

- The accuracy of the modules is tested for several human samples and is validated with various datasets, like replicated samples, nucleotide composition of the surrounding region and ribosome profiling etc. They also measured the differential IPA usage for human HeLa cells: control vs U1-AMO treated, and monocyte activated samples: treated vs untreated. In all the measures, InPACT performs very well in detecting the novel IPA events and the differential ones.
- InPACT showed significant improvement in performance compared with 2 other baselines: IPAFinder and APAIQ. Baselines are compared against the truth values using 3'-end-seq, PacBio long read and simulated samples.
- The authors also showed how InPACT can be utilized for the detection of potential IPA

events on single cell RNA-seq data.

RESPONSE: We highly appreciate your time and efforts in reviewing our manuscript and your encouraging comments.

In response to the feedback made by reviewer #1, the authors Liu et al. made substantial changes to their manuscript. They have added details about the training/testing data, including the generation of positive and negative samples, also described the sequence module and the read module in a more comprehensive way. They excluded the overlapping IPA sites from the different datasets which have been used in training and testing: RefSeq, GENCODE, PolyA_DB3 and PolyASite 2.0, to prevent the overfitting of results. Three other deep learning tools (DeepPASS, APARENT and DeepPASTA) have been mentioned and compared with the sequence module of InPACT, but no significant differences have been found in their way of selecting the annotated IPA sites from different databases. In accordance with the comment from reviewer #1, the authors also incorporated several human cell-lines and showed how InPACT outperformed the baselines (IPAFinder and APAIQ). Therefore, the quantification of the new IPA isoforms and their relative usage are also shown in the manuscript. Overall, the authors have adequately addressed most of Reviewer 1's concerns.

RESPONSE: We express our gratitude for your time and efforts in assessing our revised manuscript and responses to the comments made by reviewer #1. We sincerely appreciated the favorable evaluation of our revised manuscript.